# Multi-view light-sheet imaging and tracking with the MaMuT software reveals the cell lineage of a direct developing arthropod limb

**Carsten Wolff[1][†]\*, Jean-Yves Tinevez[2][†], Tobias Pietzsch[3][†], Evangelia Stamataki[4], Benjamin Harich[3][‡], Léo Guignard[4], Stephan Preibisch[5], Spencer Shorte[2], Philipp J Keller[4], Pavel Tomancak[3]\*, Anastasios Pavlopoulos[4]\***

[1]Institut für Biologie, Humboldt-Universität zu Berlin, Berlin, Germany; [2]Imagopole, Citech, Institut Pasteur, Paris, France; [3]Max Planck Institute of Molecular Cell Biology and Genetics, Dresden, Germany; [4]Janelia Research Campus, Howard Hughes Medical Institute, Ashburn, United States; [5]Berlin Institute for Medical Systems Biology, Max Delbrück Center for Molecular Medicine, Berlin, Germany

**\*For correspondence:**
carsten.wolff@rz.hu-berlin.de
(CW);
tomancak@mpi-cbg.de (PT);
pavlopoulosa@janelia.hhmi.org
(AP)

[†]These authors contributed
equally to this work

**Present address:** [‡]Donders
Institute for Brain Cognition and
Behaviour, Department of
Human Genetics, Radboud
University Medical Center,
Nijmegen, Netherlands

**Competing interests:** The
authors declare that no
competing interests exist.

**Reviewing editor:** Alejandro
Sánchez Alvarado, Stowers
Institute for Medical Research,
United States

**Abstract** During development, coordinated cell behaviors orchestrate tissue and organ morphogenesis. Detailed descriptions of cell lineages and behaviors provide a powerful framework to elucidate the mechanisms of morphogenesis. To study the cellular basis of limb development, we imaged transgenic fluorescently-labeled embryos from the crustacean *Parhyale hawaiensis* with multi-view light-sheet microscopy at high spatiotemporal resolution over several days of embryogenesis. The cell lineage of outgrowing thoracic limbs was reconstructed at single-cell resolution with new software called Massive Multi-view Tracker (MaMuT). In silico clonal analyses suggested that the early limb primordium becomes subdivided into anterior-posterior and dorsal-ventral compartments whose boundaries intersect at the distal tip of the growing limb. Limb-bud formation is associated with spatial modulation of cell proliferation, while limb elongation is also driven by preferential orientation of cell divisions along the proximal-distal growth axis. Cellular reconstructions were predictive of the expression patterns of limb development genes including the BMP morphogen Decapentaplegic.
DOI: https://doi.org/10.7554/eLife.34410.001

## Introduction

Morphogenesis, or the origin of biological form, is one of the oldest and most enduring problems in biology. Embryonic tissues change their size and shape during development through patterned cell activities controlled by intricate physico-chemical mechanisms (*Day and Lawrence, 2000*; *Heisenberg and Bellaïche, 2013*; *Keller, 2013*, *2012*; *Lecuit and Mahadevan, 2017*; *LeGoff and Lecuit, 2015*). Developmental processes have been explained traditionally in terms of genes and gene regulatory networks, and a major challenge is to understand how the genetic and molecular information is ultimately translated into cellular activities like proliferation, death, change of shape and movement. Therefore, detailed descriptions of cell lineages and behaviors can provide a firm ground for studying morphogenesis from a bottom-up cellular perspective (*Buckingham and Meilhac, 2011*; *Kretzschmar and Watt, 2012*; *Schnabel et al., 1997*; *Spanjaard and Junker, 2017*; *Sulston et al., 1983*).

We have focused here on the crustacean *Parhyale hawaiensis* that satisfies a number of appealing biological and technical requirements for multi-level studies of appendage (limb) morphogenesis

**eLife digest** During early life, animals develop from a single fertilized egg cell to hundreds, millions or even trillions of cells. These cells specialize to do different tasks; forming different tissues and organs like muscle, skin, lungs and liver. For more than a century, scientists have strived to understand the details of how animal cells become different and specialize, and have created many new techniques and technologies to help them achieve this goal.

Limbs – such as arms, legs and wings – form from small lumps of cells called limb buds. Scientists use the shrimp-like crustacean, *Parhyale hawaiensis*, to study development, including limb growth. This species is useful because it is easy to grow, manipulate and observe its developing young in the laboratory. Understanding how its limbs develop offers important new insights into how limbs develop in other animals too. Wolff, Tinevez, Pietzsch et al. have now combined advanced microscopy with custom computer software, called Massive Multi-view Tracker (MaMuT) to investigate this.

As limbs develop in *Parhyale,* the MaMuT software tracks how cells behave, and how they are organized. This analysis revealed that for cells to produce a limb bud, they need to split at an early stage into separate groups. These groups are organized along two body axes, one that goes from head to tail, and one that runs from back to belly. The limb grows perpendicular to these main body axes, along a new 'proximal-distal' axis that goes from nearest to furthest from the body. Wolff et al. found that the cells that contribute to the extremities of the limb divide faster than the ones that stay closer to the body. Finally, the results show that when cells in a limb divide, they mostly divide along the proximal-distal axis, producing one cell that is further from the body than the other. These cell activities may help limbs to get longer as they grow.

Notably, the groups of cells seen by Wolff et al. were expressing genes that had previously been identified in developing limbs. This helps to validate the new results and to identify which active genes control the behaviors of the analyzed cells.

These findings reveal new ways to study animal development. This approach could have many research uses and may help to link the mechanisms of cell biology to their effects. It could also contribute to new understanding of developmental and genetic conditions that affect human health.
DOI: https://doi.org/10.7554/eLife.34410.002

(*Stamataki and Pavlopoulos, 2016*). *Parhyale* is a direct developer; its body plan is specified during the 10 days of embryogenesis when imaging is readily possible (*Browne et al., 2005*). Each embryo develops a variety of specialized appendages along the anterior-posterior axis that differ in size, shape and pattern (*Martin et al., 2016*; *Pavlopoulos et al., 2009*; *Wolff and Scholtz, 2008*). *Parhyale* eggs have good size and optical properties for microscopic live imaging at cellular resolution; the eggshell is transparent and embryos are 500 μm long with low autofluorescence and light scattering. Several functional genetic approaches, embryological treatments and genomic resources also allow diverse experimental manipulations in *Parhyale* (*Kao et al., 2016*).

Previous reports have used transmitted light and fluorescence time-lapse microscopy to live image early processes like gastrulation and germband formation during the first couple days of *Parhyale* development (*Alwes et al., 2011*; *Chaw and Patel, 2012*; *Hannibal et al., 2012*). However, for a comprehensive coverage of *Parhyale* limb formation, embryos need to be imaged from multiple angular viewpoints from day 3 to day 8 of embryogenesis (*Browne et al., 2005*). We demonstrate here that transgenic embryos with fluorescently labeled nuclei can be imaged routinely for several consecutive days using Light-sheet Fluorescence Microscopy (LSFM). LSFM is an ideal technology for studying how cells form tissues and organs in intact developing embryos (*Huisken et al., 2004*; *Keller et al., 2008*; *Truong et al., 2011*). It enables biologists to capture fast and dynamic processes at very high spatiotemporal resolution, over long periods of time, and with minimal bleaching and photo-damage (*Combs and Shroff, 2017*; *Huisken and Stainier, 2009*; *Khairy and Keller, 2011*; *Schmied et al., 2014*; *Weber et al., 2014*). In addition, samples can be optically sectioned from multiple angles (multi-view LSFM) that can be combined computationally to reconstruct the entire specimen with more isotropic resolution (*Chhetri et al., 2015*; *Krzic et al., 2012*; *Schmid et al., 2013*; *Swoger et al., 2007*; *Tomer et al., 2012*; *Wu et al., 2016*; *Wu et al., 2013*).

Although the amount and type of data generated by multi-view LSFM raise several challenges for image analysis, many of them have been efficiently addressed. Software solutions exist for registration of acquired views, fusion of raw views (z-stacks) into a single output z-stack, and visualization of the raw and fused images (*Chhetri et al., 2015*; *Ingaramo et al., 2014*; *Pietzsch et al., 2015*; *Preibisch et al., 2014*; *Preibisch et al., 2010*; *Rubio-Guivernau et al., 2012*; *Wu et al., 2016*). These processes should be repeated for hundreds or thousands of time-points to generate a 4D representation of the embryo as it develops over time (*Amat et al., 2015*; *Schmied et al., 2014*; *Schmied et al., 2016*). Automated approaches for cell segmentation and tracking have also been developed (*Amat et al., 2014*; *Du et al., 2014*; *Dufour et al., 2017*; *Faure et al., 2016*; *Schiegg et al., 2015*; *Stegmaier et al., 2016*; *Ulman et al., 2017*), however they do not yet reach the precision required for unsupervised extraction of cell lineages. To address this issue, we describe here the Massive Multi-view Tracker (MaMuT) software that allows visualization, annotation, and accurate lineage reconstruction of large multi-dimensional microscopy data.

We quantitatively analyzed *Parhyale* LSFM datasets with MaMuT to understand the cellular basis of arthropod limb morphogenesis. As revealed by lineage tracing experiments in the leading arthropod model *Drosophila melanogaster*, the leg and wing primordia become progressively subdivided into distinct cell populations (called compartments when lineage-restricted) along the anterior-posterior (AP) and dorsal-ventral (DV) axes (*Dahmann et al., 2011*; *Garcia-Bellido et al., 1973*; *Steiner, 1976*). Tissue subdivisions acquire distinct cell fates driven by domain-specific expression of patterning genes (called selectors if lineally inherited), as well as by the localized induction of signaling molecules at compartment boundaries (organizers) that control patterning and growth of developing organs (*García-Bellido, 1975*; *Lawrence and Struhl, 1996*; *Mann and Carroll, 2002*; *Restrepo et al., 2014*).

Besides regionalization mechanisms, oriented cell divisions have been implicated as a general mechanism in shaping the *Drosophila* wing and other growing organs (*Baena-López et al., 2005*; *Legoff et al., 2013*; *Mao et al., 2013*). Other mechanisms like differential cell proliferation and cell rearrangement could also play a role in the formation of limb buds and their elongation along the proximal-distal (PD) axis. So far, these processes have not been possible to live image and quantify in direct developing arthropod limbs. Our understanding of cell dynamics shaping arthropod limbs has relied exclusively on studies of the indirectly developing *Drosophila* limbs (primarily the wing disc) using clonal analysis and lineage tracing across fixed specimens (*Baena-López et al., 2005*; *González-Gaitán et al., 1994*; *Resino et al., 2002*; *Weigmann and Cohen, 1999*; *Worley et al., 2013*) and recent improvements in imaging discs in vivo and ex vivo (*Dye et al., 2017*; *Heemskerk et al., 2014*; *Legoff et al., 2013*; *Mao et al., 2013*; *Strassburger et al., 2017*; *Tsao et al., 2016*; *Zartman et al., 2013*).

By tracking all constituent cells in direct developing *Parhyale* limbs, we identified the lineage restrictions and morphogenetic cellular behaviors operating during limb bud formation and elongation, and compared these to *Drosophila* and other arthropod and vertebrate paradigms. We validated our cellular models of morphogenesis by studying the expression of developmental regulatory genes implicated in limb patterning and growth.

## Results

### Imaging *Parhyale* embryogenesis with multi-view LSFM

Three-day old transgenic embryos with fluorescently labeled nuclei were mounted for LSFM in low melting agarose with scattered fluorescent beads. Several parameters were optimized to cover all stages of *Parhyale* appendage development at single-cell resolution with adequate temporal sampling for accurate cell tracking (see Materials and methods). A typical 4 to 5 day long recording was composed of more than 1 million images resulting in >7 TB datasets.

The relatively slow tempo of *Parhyale* development enabled imaging of each embryo from multiple highly overlapping views with minimal displacement of nuclei between views acquired in each time-point (*Figure 1A*). As detailed in Materials and methods, development of the entire embryo was reconstructed using the Fiji (Fiji Is Just ImageJ) biological image analysis platform (*Schindelin et al., 2012*) according to the following steps: (1) image file preprocessing, (2) bead-based spatial registration of views in each time-point, (3) fusion by multi-view deconvolution, (4)

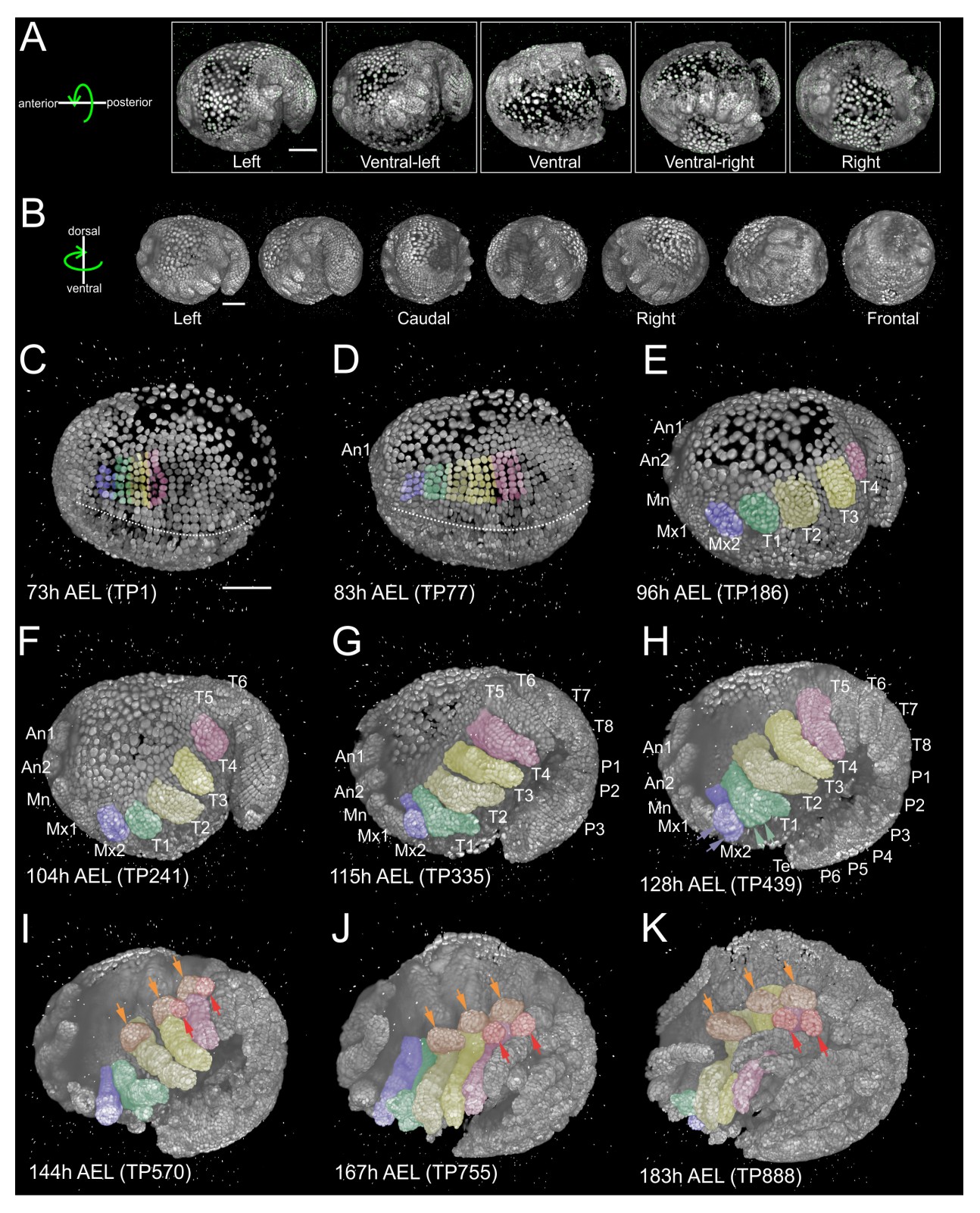

**Figure 1.** Reconstruction of *Parhyale* embryogenesis with multi-view LSFM (see also *Figure 1—video 1* and *2*). (A) Transgenic *Parhyale* embryo with H2B-mRFPruby-labeled nuclei mounted with fluorescent beads (green dots) for multi-view reconstruction. The embryo was imaged from the indicated 5 views with 45° rotation around the AP axis between neighboring views. Panels show renderings of the acquired views with anterior to the left. (B) Raw views were registered and fused into an output image rendered here in different positions around the DV axis. (C–K) Each panel shows a rendering of *Figure 1 continued on next page*

*Figure 1 continued*

the embryo at the indicated developmental stage in hours (h) after egg-lay (AEL) and the corresponding time-point (TP) of the recording. Anterior is to the left and dorsal to the top. Abbreviations: first antenna (An1), second antenna (An2), mandible (Mn), maxilla 1 (Mx1), maxilla 2 (Mx2), thoracic appendages 1 to 8 (T1–T8), pleonic (abdominal) appendages 1 to 6 (P1–P6) and telson (Te). Color masks indicate the cells contributing to Mx2 (blue), T1 (green), T2 and T3 (light and dark yellow) and T4 limb (magenta). (C) Embryo at mid-germband stage S13 according to (*Browne et al., 2005*). The ventral midline is denoted with the dotted line. (D) S15 embryo. Germband has extended to the posterior egg pole and the first antennal bud is visible anteriorly. (E) S18 embryo with posterior flexure. Head and thoracic appendages have bulged out up to T4. (F) S19 embryo with prominent head and thoracic appendage buds up to T6. (G) S20 embryo continues axial elongation ventrally and anteriorly. Appendage buds are visible up to P3. (H) S21 embryo. Segmentation is complete and all appendages have formed. The Mx2 has split into two branches (blue arrowheads) and the T1 limb has developed two proximal ventral outgrowths (green arrowheads). (I) Embryo at stage S22, (J) S23, and (K) S24 showing different phases of appendage segmentation. Dorsal outgrowths at the base of thoracic appendages, namely coxal plates (orange arrowheads) and gills (red arrowheads), are indicated in T2, T3 and T4. Scale bars are 100 µm.

DOI: https://doi.org/10.7554/eLife.34410.003

The following videos are available for figure 1:

**Figure 1—video 1.** Imaging *Parhyale* embryogenesis with multi-view LSFM

DOI: https://doi.org/10.7554/eLife.34410.004

**Figure 1—video 2.** Imaging *Parhyale* embryogenesis with multi-view LSFM

DOI: https://doi.org/10.7554/eLife.34410.005

bead-based temporal registration across time-points, (5) computation of temporally registered fused volumes, and (6) 4D rendering of the spatiotemporally registered fused data (*Preibisch et al., 2014*; *Preibisch et al., 2010*; *Schmied et al., 2014*). This processing resulted in almost isotropic resolution of fused volumes (*Figure 1B*) and was used for visualization of *Parhyale* embryogenesis with cellular resolution (*Figure 1C–K* and *Figure 1—video 1*).

Segment formation and maturation in *Parhyale* occurred sequentially in AP progression (*Figure 1—video 2*). Appendage morphogenesis involved patterning, growth and differentiation of ectodermal cells organized in an epithelial monolayer that gave rise to the appendage epidermis. In our LSFM recordings, we were particularly interested in imaging the limbs in the anterior thorax of *Parhyale* embryos that were specified at about 3.5 days after egg-lay (AEL) at 25°C. Over the next 4 days, limb buds bulged out ventrally, elongated along their PD axis and became progressively segmented until they acquired their definite morphology at around 8 days AEL (*Figure 1C–K* and *Figure 1—video 2*).

During germband formation, the ectoderm contributing to the posterior head and the trunk became organized in a stereotyped grid-like pattern with ordered AP rows and DV columns of cells (*Figure 2A–A''*) (*Browne et al., 2005*; *Dohle et al., 2004*; *Dohle and Scholtz, 1988*; *Gerberding et al., 2002*; *Scholtz, 1990*; *Wolff and Gerberding, 2015*). Each row of cells corresponded to one parasegment, which is the unit of early metameric organization in *Parhyale* embryos, like in *Drosophila* and other arthropods (*Hejnol and Scholtz, 2004*; *Scholtz et al., 1994*). Two rounds of longitudinally-oriented cell divisions in each formed parasegmental row (*Figure 2B–D'*), together with the progressive addition of new parasegments at the posterior end, led to embryo axial elongation (*Figure 1C–H*). Subsequent divisions of ectodermal cells had a more complex pattern disrupting the regularity of the grid and contributing to the transition from parasegmental to segmental body organization and the evagination of paired appendages in each segment. Appendage buds appeared successively from the head region backwards (*Figure 1D–H*) and started lengthening (*Figure 1F–K*) and differentiating along their PD axis (*Figure 1G–K*). At the end of the imaging period, morphogenesis appeared nearly complete. Thus, multi-view LSFM imaging captured the entire gamut of differential appendage morphogenetic events along the body axis of the *Parhyale* embryo in a single time-lapse experiment.

## MaMuT: a platform for cell tracking in multi-view and multi-terabyte datasets

To examine the cellular basis of morphogenesis, we developed a novel Fiji plugin to extract cell lineages from multi-view and multi-terabyte datasets. This tool was dubbed MaMuT for Massive Multi-view Tracker (*Figure 3*) and is a hybrid and extension of two existing Fiji plugins: the BigDataViewer visualization engine (*Pietzsch et al., 2015*) and the TrackMate annotation engine (*Tinevez et al., 2017*). MaMuT can be installed through the Fiji updater and is tightly integrated with the other Fiji

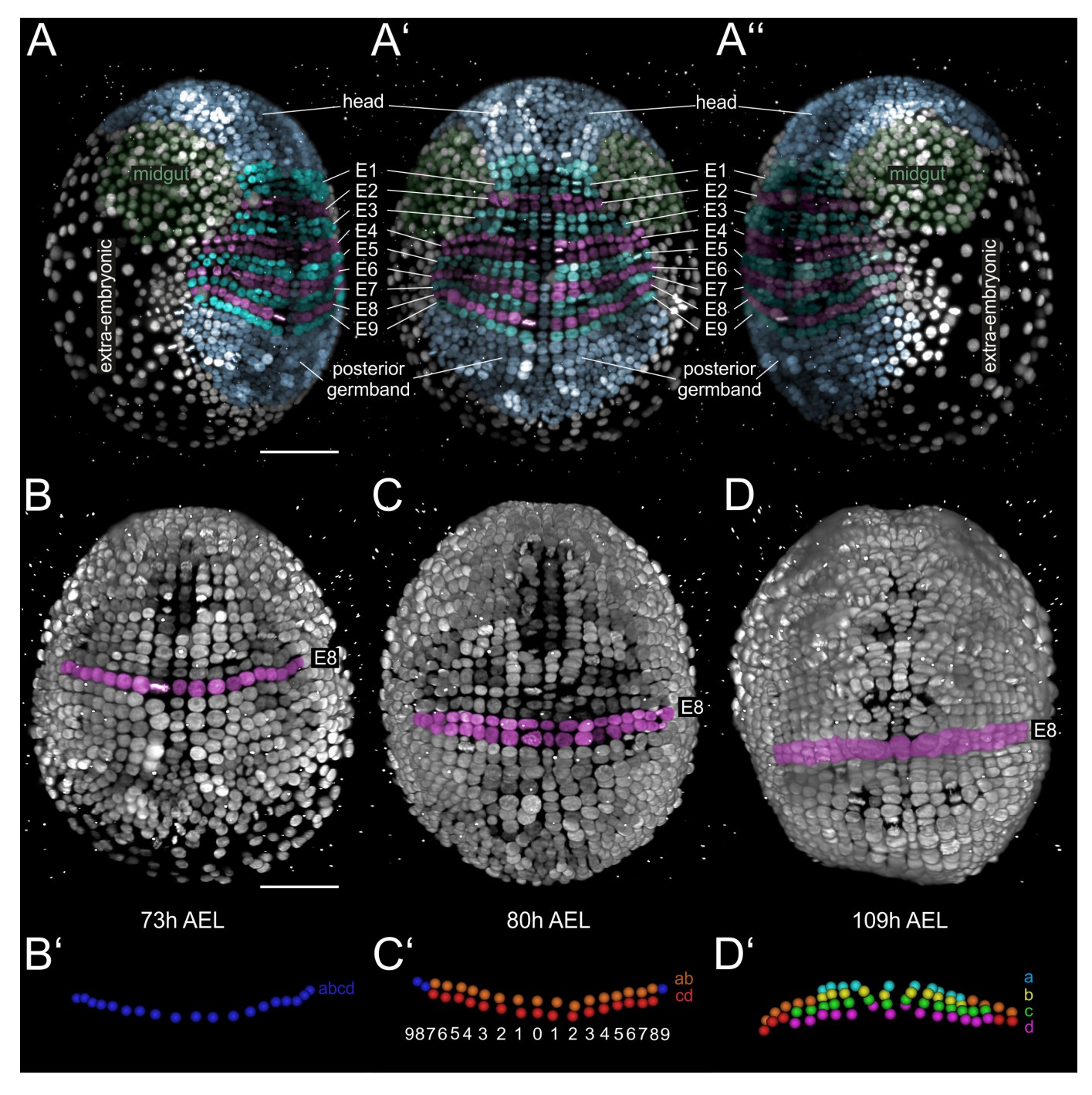

**Figure 2.** Grid architecture of the *Parhyale* germband. (A–A'') Rendering of a *Parhyale* embryo at the growing germband stage: (A) Right, (A') ventral, and (A'') left side. Color masks indicate the anterior head region (blue), the bilaterally symmetric midgut precursors (green), the orderly arranged parasegments E1 to E9 (in alternating cyan and magenta), the posterior end of the germband with ongoing organization of cells into new rows (blue), and the extra-embryonic tissue (white). (B–D) Ventral views of elongating germband at the indicated hours (h) after egg-lay (AEL). Ectodermal cells of the E8 parasegment are shown in magenta. (B') Schematics of tracked E8 abcd cells (blue) in the 1-row-parasegment, (C') anterior ab cells (orange) and posterior cd cells (red) after the first longitudinally-oriented division in the 2-row-parasegment, and (D') a (cyan), b (yellow), c (green) and d cells (magenta) after the second longitudinally-oriented division in the 4-row-parasegment. Both mitotic waves proceed in medial-to-lateral direction. The resulting daughter cells sort in clearly defined columns that are identified by ascending index numbers with 0 denoting the ventral midline and 1, 2, 3 etc. the more lateral columns with increasing distance from midline.

DOI: https://doi.org/10.7554/eLife.34410.006

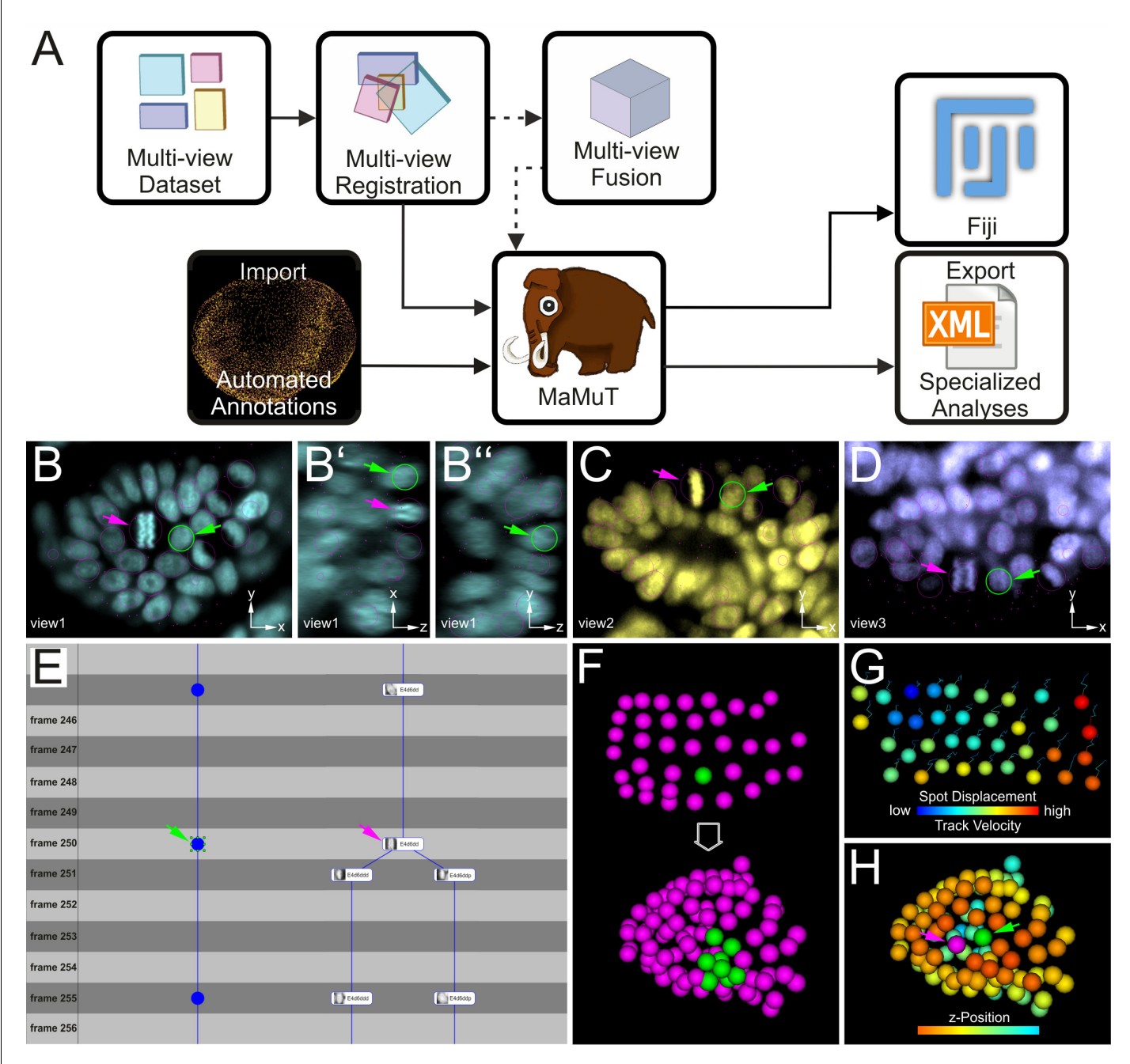

**Figure 3.** Cell tracking and lineage reconstruction with MaMuT (see also *Figure 3—figure supplement 1*). (A) Workflow for image data analysis with MaMuT. Raw views (colored boxes in Multi-view Dataset) are registered (overlapping boxes in Multi-view Registration) and, optionally, fused into a single volume (large cube in Multi-view Fusion). The raw (and fused) image data together with the registration parameters are imported into MaMuT (mammoth logo). In its simplest implementation, all data analysis is done with MaMuT in Fiji workspace. In more advanced implementations, automated segmentation and tracking annotations (yellow point cloud of tracked cells) can be computed separately and imported into MaMuT. The reconstructed lineage information can be exported from MaMuT in an xml file for specialized analyses in other platforms. (B–D) The MaMuT Viewer windows display the raw image data and annotations. All tracked nuclei are marked with magenta circles (in view) or dots (out of view). The active selection is marked in green in all synced Viewers: (B) xy, (B') xz, and (B'') yz plane of first view in cyan; (C) xy plane of second view in yellow; (D) xy plane of third view in blue. (E) The TrackScheme lineage browser and editor where tracks are arranged horizontally and time-points vertically. Tracked objects can be displayed simply as spots (left track) or with extra information like their names and thumbnails (right track). Tracks are displayed as vertical links. The TrackScheme is synced with the Viewer windows; the selected nucleus in panels B–D is also highlighted here in green at the indicated time-point (called frame). Objects can be tracked between consecutive time-points or in larger steps. (F–H) The 3D Viewer window displays interactive animations of tracked objects depicted as spheres. Spots and tracks can be color-coded by lineage, position and other numerical parameters extracted from the data. (F)

*Figure 3 continued on next page*

*Figure 3 continued*

Digital clone of a nucleus (shown in green) tracked from the grid stage to the limb bud stage. All other tracked nuclei are shown in magenta. (G) Spots color-coded by displacement and tracks color-coded by velocity. (H) Tracked nuclei in the limb bud mapped out in different colors based on z-position. In panels B–E and H, the selected nucleus and the neighboring dividing nucleus are indicated with green and magenta arrowheads, respectively.

DOI: https://doi.org/10.7554/eLife.34410.007

The following figure supplement is available for figure 3:

**Figure supplement 1.** MaMuT layout.

DOI: https://doi.org/10.7554/eLife.34410.008

plugins for LSFM data processing. The source code for MaMuT is available on GitHub (*Tinevez et al., 2018*; copy archived at https://github.com/elifesciences-publications/MaMuT) and detailed tutorials and training datasets can be found at http://imagej.net/MaMuT.

MaMuT is an interactive, user-friendly tool for visualization, annotation, tracking and lineage reconstruction of large multi-dimensional microscopy data (*Figure 3* and *Figure 3—figure supplement 1*). It is a versatile platform that can be used either for manual or semi-automated tracking of selected populations of cells of interest, or for visualization and editing of fully automated computational predictions for systems-wide lineage reconstructions. MaMuT can handle multiple data sources but was developed primarily to enable the analysis of LSFM datasets. Its unique feature is the ability to annotate image volumes synergistically from all available input views (detailed in Materials and methods). This functionality of MaMuT allowed us to identify and track all constituent cells in developing limbs continuously from the early germband stages until the later stages of 3D organ outgrowth, when the information from multiple views was required for full reconstructions.

## Single-cell lineage reconstruction of a *Parhyale* thoracic limb

We deployed the manual version of MaMuT to extract the lineage of one *Parhyale* thoracic limb. By convention, *Parhyale* parasegments are identified by ascending indices E1, E2, E3 etc., the AP rows of ectodermal cells in each parasegment by the letters a, b, c and d, and the DV columns of cells in each parasegment by numbers (*Figure 2*). In accordance with previous studies in malacostracan crustaceans and other arthropods, our reconstructions demonstrated that each *Parhyale* thoracic limb consisted of cells from two neighboring parasegments (*Browne et al., 2005*; *Dohle et al., 2004*; *Dohle and Scholtz, 1988*; *Hejnol and Scholtz, 2004*; *Scholtz, 1990*; *Scholtz et al., 1994*; *Wolff and Scholtz, 2008*). The T2 limb (referred to as limb#1) that we analyzed in-depth (*Figure 4A–E*) developed from rows b, c and d of the E4 parasegment and from rows a and b of the following E5 parasegment (*Figure 4F–J'*). Cells that arose from rows c, d and a occupied the entire length of the limb and body wall parts of the T2 segment, while rows b contributed only to the proximal limb and intersegmental territories (*Figure 4—figure supplement 1K–O'*). Cells in medial columns 1 and 2 gave rise to the nervous system and sternites and were not considered in this study. The more lateral columns 3 to 9 gave rise to the forming limb (*Figure 4—figure supplement 1F–J*).

We fully tracked 34 founder cells constituting the limb#1 primordium over 50 hr of development, giving rise to a total of 361 epidermal cells (*Figure 5* and *Figure 5—video 1*). We started tracking each of these 34 cells as they were born during transition from the 2-row to the 4-row-parasegment (*Figure 5A–C*), and then continuously during the subsequent rounds of divisions, referred to as differential divisions (DDs) (*Figure 5D*). The number of DDs observed during these 50 hr varied dramatically between cells from just 1 DD in the slowest dividing lateral cells of the primordium (cells E4b8, E5a9/b9) to 5 DDs in the fastest dividing central cells (cells E4c3-c6 and E4d3-d6). Although the clonal composition of crustacean appendages had been described previously with lipophilic dye injections (*Wolff and Scholtz, 2008*), the reconstruction presented here is the most comprehensive lineage tree for any developing arthropod limb published to date (*Figure 5—figure supplement 1*).

## Early lineage restrictions along the AP and DV axes

We first asked whether these complete reconstructions could reveal any lineage-based subdivisions in the developing limb#1. The AP restriction at the border of neighboring parasegments at the 1-row stage has been revealed in *Parhyale* and other embryos by embryological descriptions, lineage tracing and expression studies for the *engrailed* (*en*) gene that marks the posterior compartment

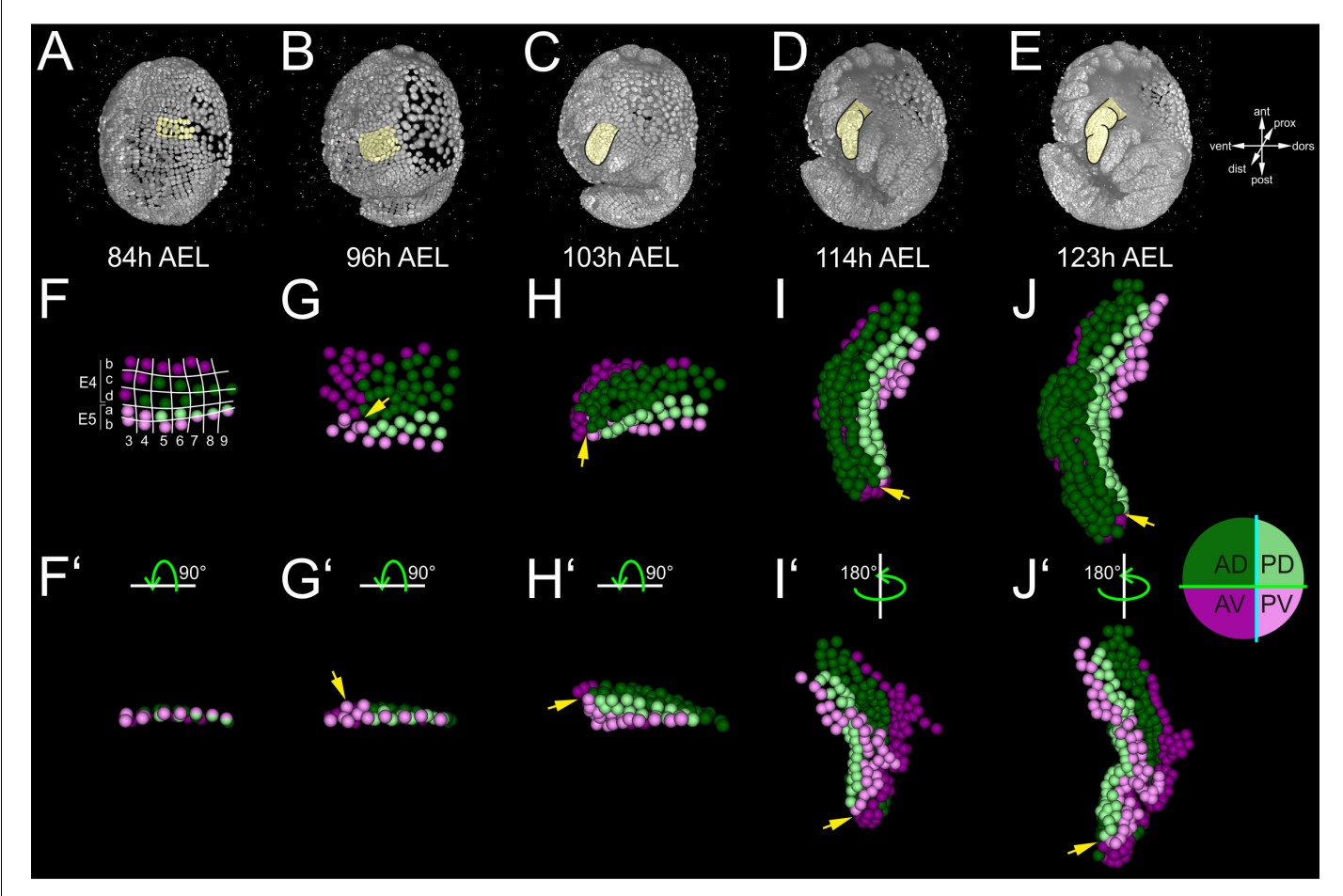

**Figure 4.** Early compartmentalization of the *Parhyale* thoracic limb (see also *Figure 4—figure supplements 1* and *2*). (A–E) Lateral views of a *Parhyale* embryo rendered at the indicated developmental stages shown in hours (h) after egg-lay (AEL). Yellow masks show the left T2 limb (limb#1). (F–J') Tracked cells contributing to limb#1 were color-coded by their compartmental identity: Anterior-Dorsal (dark green), Anterior-Ventral (dark magenta), Posterior-Dorsal (light green), and Posterior-Ventral (light magenta). (F) Ventral view of limb primordium at 84 hr AEL made up by cells from the E4 and E5 parasegments. Horizontal lines separate AP rows a to d and vertical lines separate DV columns 3 to 9. (F') Posterior view, rotated 90° relative to F. (G) Ventral view of the limb during early eversion at 96 hr AEL. (G') Posterior view, rotated 90° relative to G. The cells close to the intersection of the four compartments (yellow arrows) are the first to rise above the level of the epithelium. (H–J) Dorsal views of (H) limb bud at 103 hr AEL, (I) initial limb elongation at 114 hr AEL and (J) later elongation phase at 123 hr AEL. (H') Posterior view, rotated 90° relative to H, and (I'–J') ventral views, rotated 180° relative to I-J. The intersection of the AP and DV boundaries (yellow arrows) is located at the tip of the limb.
DOI: https://doi.org/10.7554/eLife.34410.009

The following figure supplements are available for figure 4:

**Figure supplement 1.** Lineage reconstruction of the *Parhyale* thoracic limb.
DOI: https://doi.org/10.7554/eLife.34410.010
**Figure supplement 2.** Independent evidence for early compartmentalization of the *Parhyale* thoracic limb.
DOI: https://doi.org/10.7554/eLife.34410.011

(*Browne et al., 2005*; *Dohle et al., 2004*; *Dohle and Scholtz, 1988*; *Hejnol and Scholtz, 2004*; *Patel et al., 1989*; *Scholtz, 1990*; *Scholtz et al., 1994*). In agreement with this AP restriction, during limb specification and outgrowth there was a straight clonal boundary running between the anterior cells derived from the E4b, c and d rows and the posterior cells derived from the E5a and b rows (*Figure 4F–J'* and *Figure 4—figure supplement 1K–O'*).

After the well-known AP boundary, we sought to identify any subdivision along the DV axis. Compartments were classically discovered by clonal analysis using mitotic recombination. In our reconstructions, we could generate clones digitally from arbitrary cells at different stages of development.

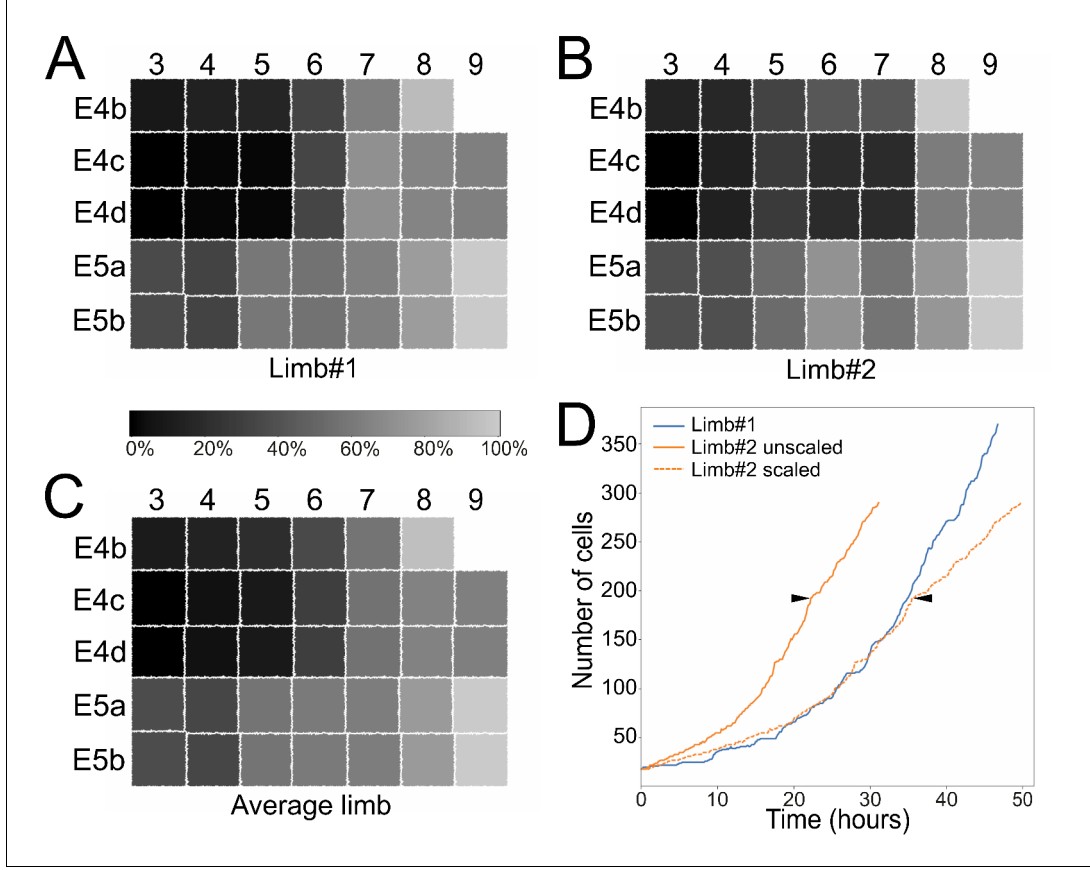

**Figure 5.** Stereotyped and variable cell behaviors in developing *Parhyale* thoracic limbs (see also *Figure 5—figure supplement 1*, *Figure 5—source data 1*, and *Figure 5—video 1*). (**A–C**) Schematic representations of the T2 limb primordium at the 4-row-parasegment stage displaying the 34 founder cells as squares color-coded based on their relative birth times: (**A**) limb#1, (**B**) limb#2 and (**C**) their average. The first forming E4c3/d3 cells are colored in black (0% birth time difference), the last forming E5a9/b9 cells in light gray (100% birth time difference) and all other cells in intermediate grayscale shades based on their birth time difference relative to E4c3/d3. (**D**) Change in cell number over time in limb#1 (blue line) imaged at 26°C and in the faster developing limb#2 (orange lines) imaged at 29–30°C. The first division of the E4cd3 cell is the starting point for both growth curves. Solid lines show the raw data for the two limbs, while the dashed orange line shows the temporally registered data for limb#2. Arrowheads indicate the unscaled and scaled time-point up to which the SIMI°BioCell reconstruction of limb#2 was complete. An increasing number of cells in limb#2 were not possible to track after this time-point resulting in a poor registration with the growth curve of limb#1.

DOI: https://doi.org/10.7554/eLife.34410.012

The following video, source data, and figure supplement are available for figure 5:

**Source data 1.** Relative birth times of founder cells in Parhyale thoracic limbs.
DOI: https://doi.org/10.7554/eLife.34410.013

**Figure supplement 1.** Reconstructed lineage tree of a *Parhyale* T2 limb.
DOI: https://doi.org/10.7554/eLife.34410.014

**Figure 5—video 1.** Animation of tracked cells forming the *Parhyale* second thoracic limb
DOI: https://doi.org/10.7554/eLife.34410.015

We reasoned that we could reveal the timing and position of any heritable DV restriction by piecing together correctly all founder cells of dorsal or ventral identity in a way that the two polyclones (i.e. compartments) would stay separate and form a lasting straight interface between them. This analysis suggested that there is indeed a DV separation that took place at the 4-row-parasegment. The DV boundary ran between the E4b and c rows anteriorly, between the E5a and b rows posteriorly, and between cells E4c4-c5, E4d3-d4 and E5a4-a5 medially (*Figure 4F*). Throughout limb#1 development,

the dorsal and ventral cells formed a sharp boundary between themselves extending along the PD axis (*Figure 4F–J'*).

To investigate the stereotypy of the AP and DV separation across *Parhyale* limbs, we analyzed a second, independently imaged and reconstructed T2 limb (referred to as limb#2) from a different embryo (*Figure 4—figure supplement 2A–D*). Four identical compartments (anterior-dorsal, anterior-ventral, posterior-dorsal and posterior-ventral) could be derived in this independent reconstruction with straight boundaries and no cell mixing between neighboring compartments (*Figure 4— figure supplement 2E*–H'). These results suggested that in silico studies of comprehensive and accurate lineages can provide novel insights into clonal subdivisions in species where sophisticated genetic methodologies for lineage tracing are not implemented yet.

## Cellular dynamics underlying limb morphogenesis

The first T2 limb (limb#1) was lineaged with the new MaMuT software from a multi-view acquisition of an embryo imaged at 26°C (*Figure 4*), while the second T2 limb (limb#2) was lineaged with the previously developed SIMI°BioCell software (*Hejnol and Scholtz, 2004*; *Schnabel et al., 1997*) from a single-view of another embryo imaged at 29–30°C (*Figure 4—figure supplement 2*). Analysis of the birth sequence of the founder cells in the two reconstructed T2 limbs largely confirmed that the second mitotic wave creating the 4-row-parasegment propagated from anterior to posterior rows and from medial to lateral columns (*Figure 5A–C* and *Figure 5—source data 1*). For example, division of the ab cells in parasegment E4 had already progressed to column 5 or even more laterally before ab3 divided in the next posterior parasegment E5. However, we also found two notable deviations from this general pattern. First, as previously noted (*Scholtz, 1990*), division of the posterior cd cells within the 2-row-parasegment was slightly more advanced temporally compared to their anterior ab sister cells (*Figure 5A–C*). Second, the temporal sequence of divisions, which gave rise to a stereotyped number and spatial arrangement of the 34 founder cells in each primordium, exhibited a certain degree of variability between the two analyzed limbs; for example, division of the E4cd8/9 cells preceded division of E4cd7 in limb#1 but not in lim#2, whereas division of the E4cd6/7 cells preceded division of E4cd5 in limb#2 but not in limb#1 (*Figure 5A–B*).

We then examined the increase in cell number over time in the two limbs during the analyzed stages of limb outgrowth. The embryo with limb#2 imaged at higher temperature exhibited a faster growth rate compared to the embryo with limb#1 (*Figure 5D*). Yet, it was possible to register the two growth curves during the period when all cells were tracked faithfully by applying a linear temporal rescaling factor of 1.6, effectively correcting for the temperature-induced change in growth (*Figure 5D*). After this temporal alignment, the increase in cell number was very similar between developing limbs, up to 35 hr after the first tracked division. The matching curves demonstrated that cell numbers were highly reproducible between developing limbs after aligning them temporally and allowed their pairwise quantitative comparison (see next section). Beyond this time-point, it was not possible to track all cells in the outgrowing limb#2 due to their increasing higher density, the deterioration of the fluorescence signal along the detection axis and the lack of the multi-view information for lineaging this limb.

Limb bud formation entailed the remodeling of the flat epithelium into a 3D bulge (*Figure 4A–C* and *Figure 4—figure supplement 2A–C*). At the cellular level, the first step in this transformation was the rise of few cells at the intersection of the four compartments above the level of the germband at around 96 hr AEL (*Figure 4G,G'* and *Figure 4—figure supplement 2F,F'*). Within the following 3 hr, this initial phase was followed by a large-scale elevation of most cells in the dorsal compartment. As this elevation continued, the medial ventral cells folded and became apposed to the medial dorsal cells forming the convex surface of the limb bud (*Figure 4H,H'* and *Figure 4—figure supplement 2G,G'*). The intersection of the AP and DV boundaries was at the tip of the limb bud and persisted in this position throughout subsequent elongation (*Figure 4H–J'* and *Figure 4— figure supplement 2G*–H'). From 103 hr AEL onwards, a second element appeared bulging distally off the original bud in limb#1 (*Figure 4I,I'*). The limb elongated as a convoluted rather than straight cylinder and acquired progressively an S-shape (*Figure 4J,J'*).

## Quantification of differential cell behaviors during limb bud formation and elongation

Two cell behaviors implicated in organ morphogenesis were readily quantifiable in our nuclear trackings: the pattern of cell proliferation and the orientation of cell divisions. These cell activities have been traditionally inferred from the distribution, size and shape of somatic clones induced in developing tissues (*Baena-López et al., 2005*; *González-Gaitán et al., 1994*; *Mao et al., 2013*; *Resino et al., 2002*; *Weigmann and Cohen, 1999*; *Worley et al., 2013*). This approach could be also adapted here by generating in silico clones (*Figure 6—figure supplement 1*). Yet, the MaMuT reconstructions enabled us to enrich the lineage information with rigorous quantitative analyses of the rate and orientation of mitotic divisions for all tracked nuclei.

First, we calculated the cell cycle length (CCL), i.e. the branch length for every constituent cell in the lineage of limb#1 (*Figure 6A–D* and *Figure 6—figure supplement 2*). This quantification revealed a striking difference in CCL between central cells that were dividing faster than their neighbors in the periphery of the primordium (average CCL 7.1–8.5 hr versus 8.5–16.4 hr, respectively). This difference started from early primordium specification at the 4-row-parasegment (*Figure 6E*), but became most pronounced during the global elevation of the limb bud (*Figure 4F*), suggesting a causal association between spatially controlled cell proliferation and initiation of limb outgrowth (see Discussion). During subsequent elongation stages, a high concentration of fast dividing cells was located at the intersection of the four presumptive compartments, resembling a growth zone at the

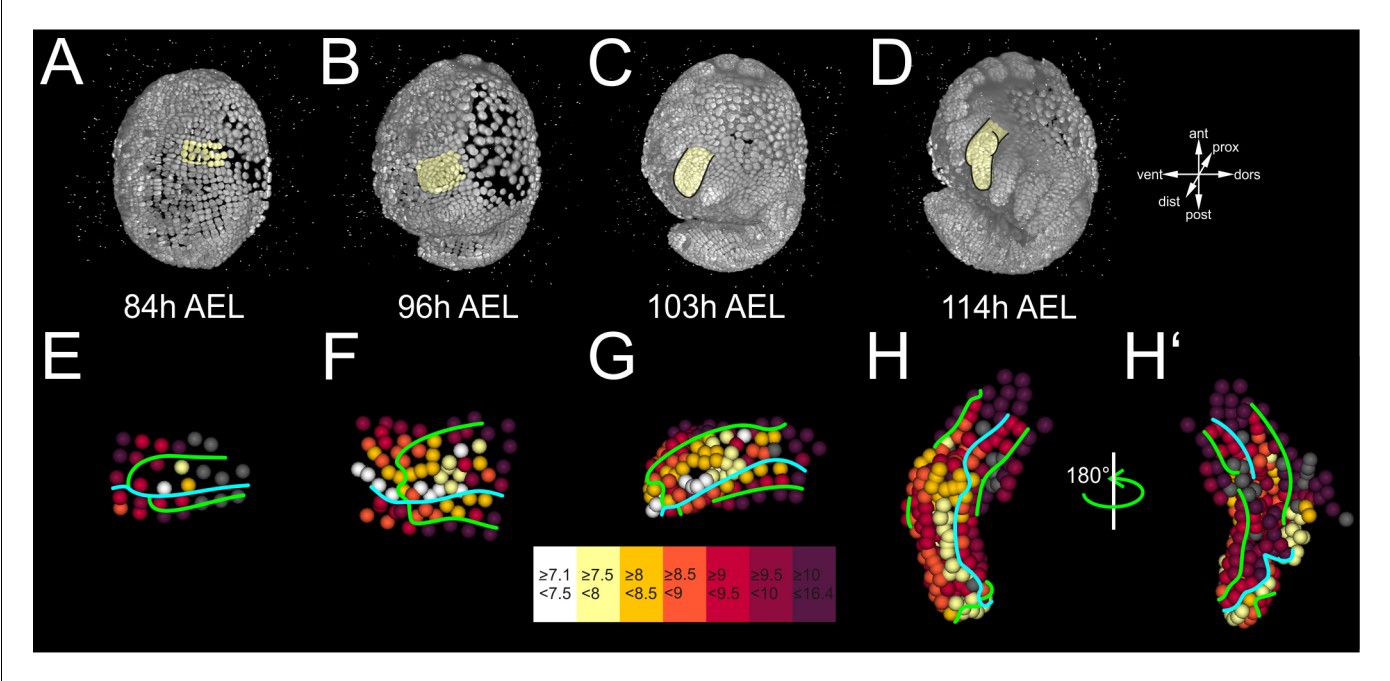

**Figure 6.** Differential cell proliferation rates in the *Parhyale* thoracic limb (see also *Figure 6—figure supplements 1* and *2*). (A–D) Lateral views of the same *Parhyale* embryo shown in *Figure 4*. (E–H') Tracked cells in limb#1 were color-coded by their average cell cycle length according to the scale (in hours) shown at the bottom. AP and DV boundaries are indicated by the cyan and green line, respectively. Cells for which measurements are not applicable are shown in gray. (E) Ventral view of the limb primordium at 84 hr (h) after egg-lay (AEL). Some central c and d cells start dividing faster at the 4-row-parasegment. (F) Ventral view of the limb during early eversion at 96 hr AEL with the middle cells dividing faster than peripheral cells. (G) Dorsal view of limb bud at 103 hr AEL. Higher proliferation rates are detected at the tip and in the anterior-dorsal compartment. (H) Dorsal and (H') ventral view of elongating limb at 114 hr AEL. Cells at the tip of the limb and anterior cells abutting the AP compartment boundary divide the fastest.

DOI: https://doi.org/10.7554/eLife.34410.016

The following figure supplements are available for figure 6:

**Figure supplement 1.** Digital clonal analysis in the *Parhyale* thoracic limb.

DOI: https://doi.org/10.7554/eLife.34410.017

**Figure supplement 2.** Alternative quantifications of cell proliferation rates in the *Parhyale* thoracic limb.

DOI: https://doi.org/10.7554/eLife.34410.018

distal tip of the growing appendage (*Figure 6G,H*). Another row of faster dividing cells was localized in the anterior cells abutting the AP boundary (*Figure 6H,H'*).

To explore the levels of variability in the pattern of cell divisions, we performed a hierarchical clustering of the founder cells within each of the two analyzed T2 limbs based on a lineage distance metric computed from the division patterns exhibited by the 34 cells (see Material and methods). This analysis revealed very similar profiles in the two limbs, as well as their average, with their cells forming three clusters (*Figure 7A–C* and *Figure 7—source data 1*): the first cluster contained cells E4c3-c7 and E4d3-d7 displaying the fastest proliferation rates and giving rise to most of the limb structures; the second cluster contained the majority of E4 and E5 b cells corresponding to the slowest dividing cells of ventral fate and contributing to the proximal limb and intersegmental territories; the third cluster contained the remaining cells exhibiting mixed division patterns, including most of the posterior E5a cells and the more lateral E4c and d cells. This clustering suggested that a common set of patterning mechanisms operates across T2 limbs specifying the distinct properties of these groups of cells. At the same time, the linkages and distances of cells within each cluster varied from identical (e.g. E4b3/b4) to very different (e.g. E4d4/d5) between limbs, revealing a certain degree of flexibility in the behaviors exhibited by homologous cells in a limb-specific manner. Extra support for this interpretation came from plotting the distribution of the lineage distances between founder cells across the two limbs. Pairwise comparisons revealed low distances between the 34 homologous cells in limb#1 and limb#2 with a median difference of 19.3% (*Figure 7D*). Thus, homologous cells exhibited similar but not identical division patterns across limbs. The distribution of these distances between homologous cells was significantly shifted towards lower values relative to pairwise

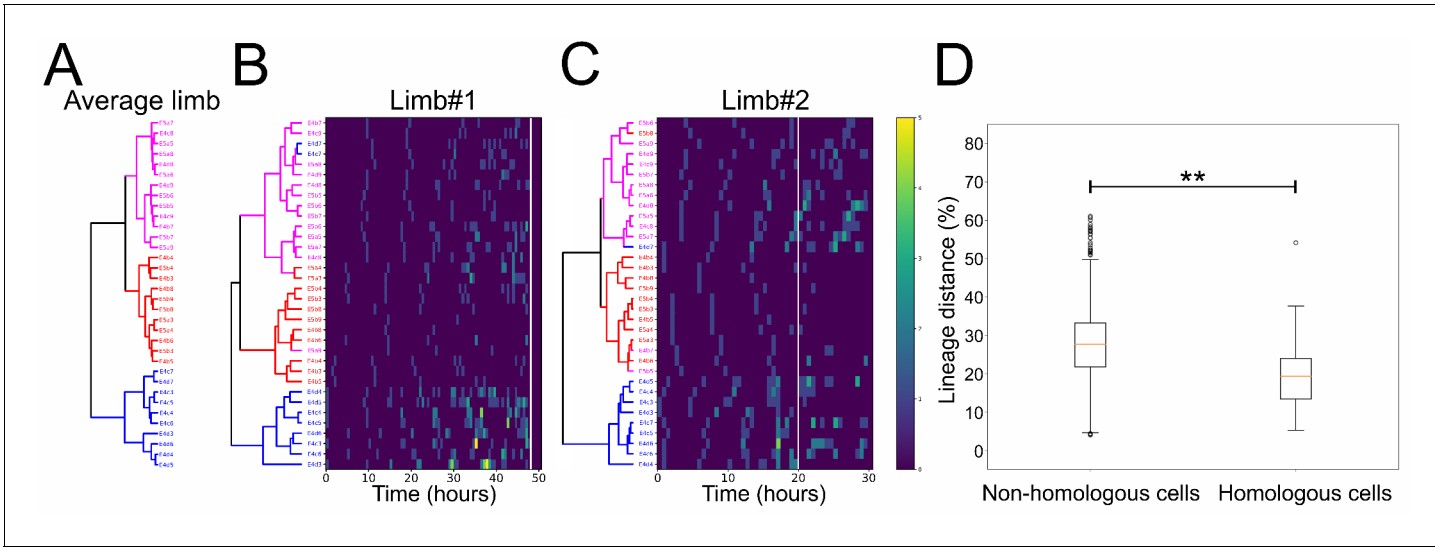

**Figure 7.** Lineage comparisons within and across *Parhyale* thoracic limbs (see also *Figure 7—source data 1*). (A) Hierarchical clustering of the 34 founder cells in the *Parhyale* T2 limb based on a distance matrix computed from their average division patterns in limb#1 and limb#2. The cluster of E4c3-c7 and E4d3-d7 cells at the bottom is shown in blue, the middle cluster containing primarily the E4 and E5 b cells is shown in red, and the top cluster with the remaining cells is shown in magenta. (B,C) Hierarchical clustering of the 34 founder cells in (B) limb#1 and (C) limb#2 based on distance matrices computed from the division patterns observed in each limb. The cells in the two trees (color-coded as in A) display very similar clustering profiles. Heat maps show the timing and number of divisions in five time-point-windows. For each founder cell, divisions are represented with rectangles color-coded according to the number of divisions shown in the color bar. The x-axis shows the unscaled tracking time for each limb starting from division of the E4cd3 cell and the white line indicates the time-point at which cells were compared. (D) Box plots showing the distribution of lineage distances in pairwise comparisons between non-homologous (left) and homologous (right) founder cells across the two limbs. The two distributions differ significantly at p≤0.01 based on the Kolmogorov-Smirnov test. The data used for limbs #1 and #2 in panels A and D were at a comparable stage of their development indicated with the arrowheads in *Figure 5D*.

DOI: https://doi.org/10.7554/eLife.34410.019

The following source data is available for figure 7:

**Source data 1.** Lineage distances between founder cells in Parhyale thoracic limbs.
DOI: https://doi.org/10.7554/eLife.34410.020

comparisons between non-homologous cells across the two limbs (*Figure 7D* and *Figure 7—source data 1*).

Next, we looked for any biases in the orientation of mitotic divisions that could be associated with limb morphogenesis (*Figure 8A–E*). All early divisions in the limb#1 primordium were parallel to the AP axis confirming the strict longitudinal orientation of row divisions (*Figure 8F*). Cell divisions acquired a more heterogeneous pattern after the 4-row-parasegment (*Figure 8G*). An increasing number of mitotic spindles aligned progressively along the PD axis during limb bud formation (*Figure 8H*) and elongation (*Figure 8I,J*). Collectively, the information extracted from our spatiotemporally resolved lineage trees strongly suggested that *Parhyale* limb outgrowth is driven by at least two patterned cell behaviors: the differential rates of cell proliferation and the orderly arrangement of mitotic spindles.

## Cellular basis of the elaboration of the limb PD axis

To understand the cellular basis of the establishment of positional values along the PD axis, we followed the fate of cells during T2 limb#1 segmentation. Segmentation involved the progressive subdivision of the elongating PD axis into an increasing number of elements (*Figure 9A–L*). We tracked neighboring cells in rows E4c (cells E4c5-c8, not shown) and E5a (cells E5a5-a8, shown in *Figure 9A–F*) from 84 to 151 hr AEL. These cells were ideal for reconstructing the PD axis at single-cell resolution because they mostly divided proximodistally forming elongated thin clones (*Figure 6—figure supplement 1*).

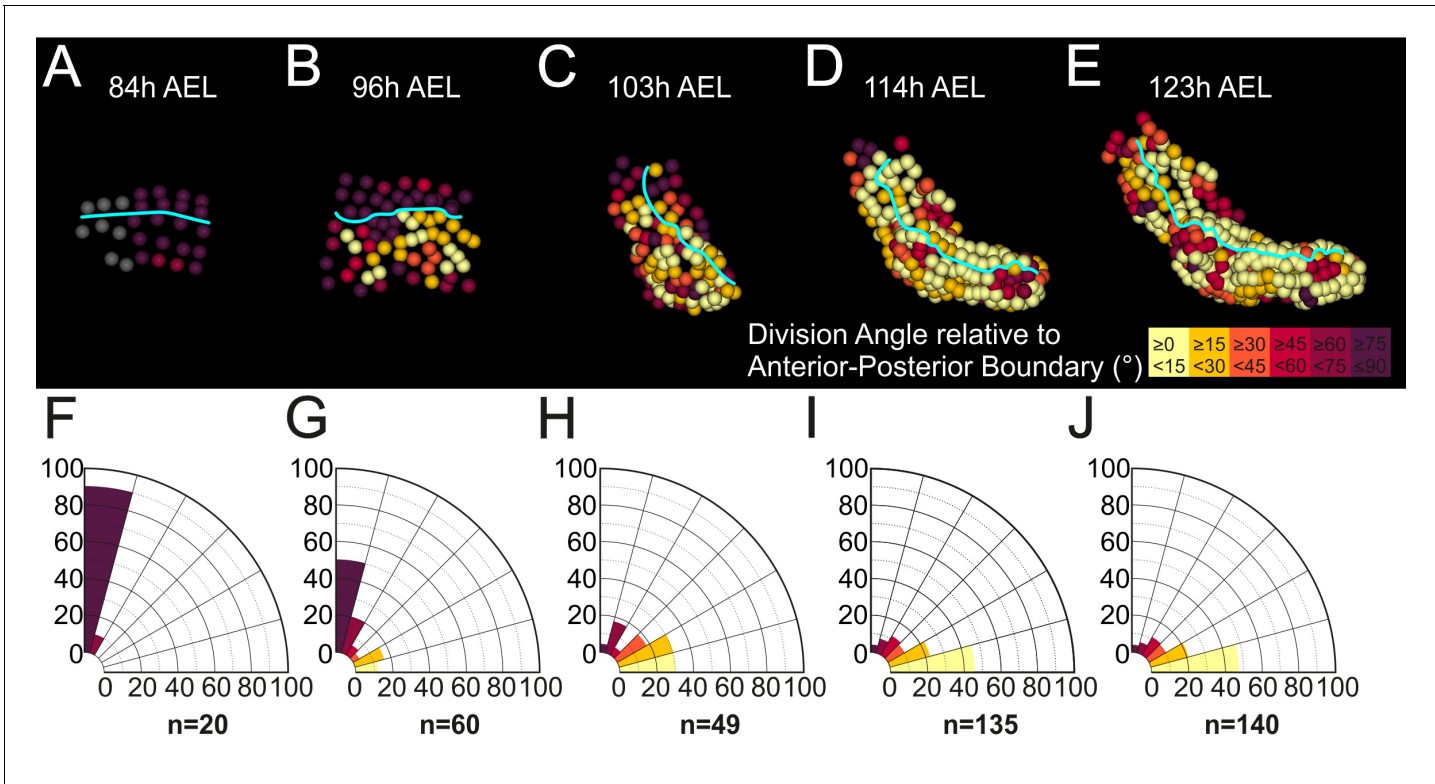

**Figure 8.** Oriented cell divisions in the *Parhyale* thoracic limb. (**A–E**) Cells in the T2 limb#1 shown at the indicated hours (h) after egg-lay (AEL) color-coded by the orientation of mitotic divisions relative to the AP boundary (cyan line). The AP boundary is parallel to and an accurate proxy for the PD axis during limb outgrowth. The absolute values of the division angle relative to the AP boundary are sorted in 6 bins of 15°. Gray cells in panel A indicate non-divided cells. (**F–J**) Rose diagrams with 15° intervals showing the percentage of mitotic events falling in each bin color-coded as in A-E (n shows the actual number of divisions). (**A,F**) Only longitudinally-oriented divisions (perpendicular to the AP boundary) are detected in the limb primordium 73 to 84 hr AEL. (**B,G**) Most cells still divide longitudinally 84 to 96 hr AEL, but an increasing number of dividing cells align parallel to the AP boundary during early eversion. (**C,H**) More than 59% of cells divide 0°−30° relative to the AP boundary in the limb bud from 96 to 103 hr AEL. (**D,I**) Early and (**E,J**) later limb elongation phase from 103 to 123 hr AEL with the large majority of cells (>68%) dividing 0°−30° relative to the AP boundary.
DOI: https://doi.org/10.7554/eLife.34410.021

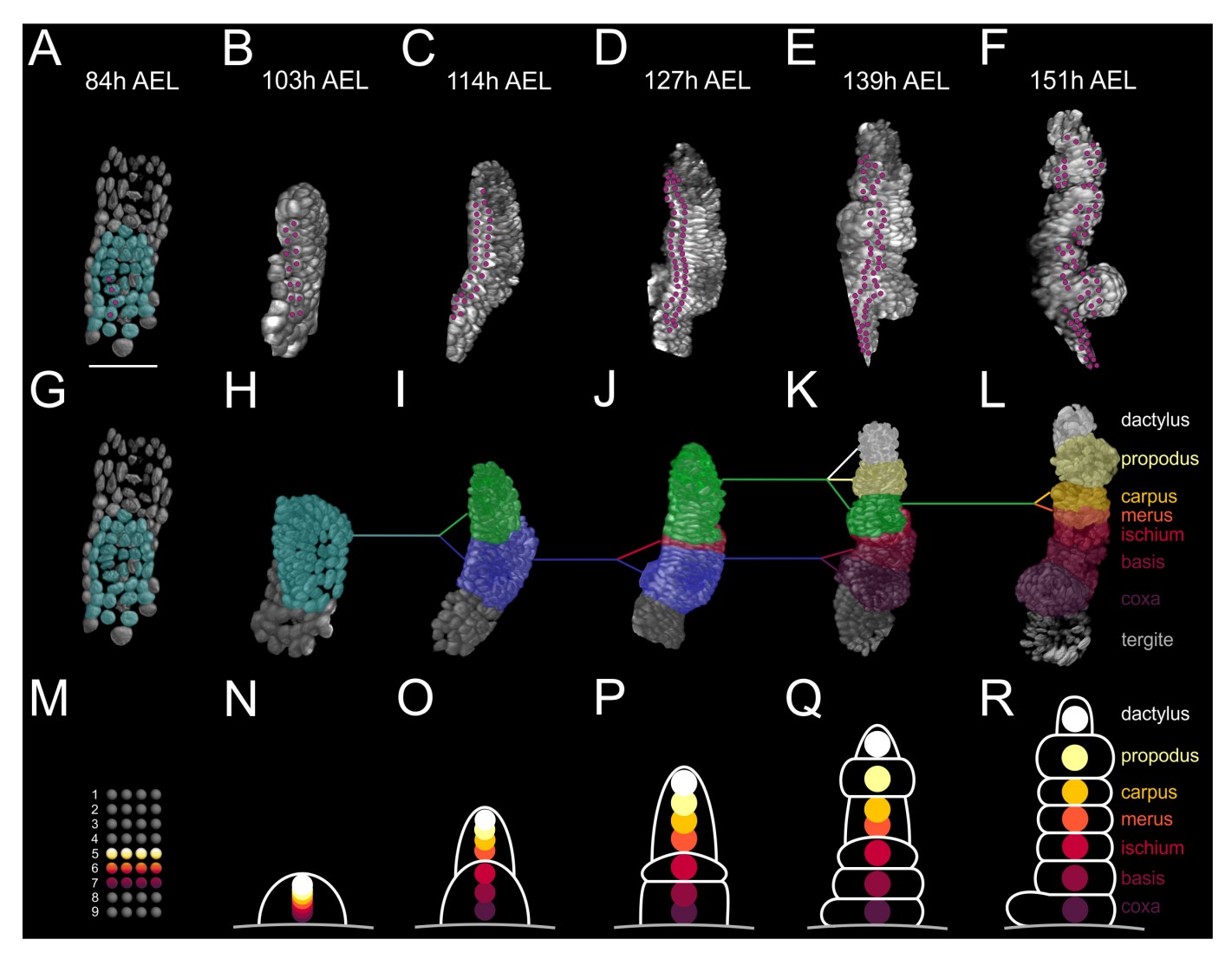

**Figure 9.** Elaboration of the *Parhyale* limb PD axis (see also *Figure 9—figure supplement 1*). (A–F) Rendering of the T2 limb#1 at the indicated hours (h) after egg-lay (AEL). The cells contributing to the T2 primordium are shown in cyan in panel A. Magenta dots indicate the tracked cells E5a5-a8 and their descendants. Panel A shows a ventral view of the germband and panels B–F posterior views of the T2 limb. (G–L) Same stages as in A–F with color masks showing (G) the limb primordium, (H) the early limb bud, (I) the 2-partite limb with the first subdivision between ischium/merus, (J) the 3-partite limb after the second subdivision between basis/ischium, (K) the 6-partite limb after three more subdivisions between coxa/basis, propodus/dactylus and carpus/propodus, and (L) the final pattern made of 7 segments after the carpus/merus division. Colored lines indicate the relationships between limb parts in consecutive stages. (M–R) Schematics of limb subdivisions along the PD axis at the same time-points as in panels G–L. The rectangular lattice in panel M shows the 9 columns of cells in the 4-row-parasegment. White lines in panels N–R delineate the subdivisions of the T2 limb. The origin of each of the seven limb segments is shown with discs color-coded by segment.

DOI: https://doi.org/10.7554/eLife.34410.022

The following figure supplement is available for figure 9:

**Figure supplement 1.** Proximal-distal lineage separation in the growing *Parhyale* thoracic limb.
DOI: https://doi.org/10.7554/eLife.34410.023

This analysis showed that the cells that gave rise to the proximal, medial and distal limb segments occupied distinct mediolateral positions in the germband grid at the 4-row-parasegment (*Figure 9M*) and distinct PD positions in the early limb bud (*Figure 9N*). When the limb bud split into two elements, the proximal element gave rise to the proximal limb segments coxa, basis and ischium, while the distal element gave rise to the distal limb segments merus, carpus, propodus, and dactylus (*Figure 9O–R*). The cells forming the distal segments originated as a disc of cells centered

at the intersection of the four compartments with contributions from the E4c4-c6, E4d3-d6 and E5a3-a6 sublineages (*Figure 9—figure supplement 1*). During the subsequent elongation stages, distal cells kept separate from more proximal cells at the prospective ischium/merus joint, suggesting that limb segments may pose secondary lineage restrictions along the PD axis (*Figure 9—figure supplement 1*) (*Milán and Cohen, 2000*). This first ischium/merus subdivision (*Figure 9O*) was followed by the basis/ischium subdivision (*Figure 9P*), the propodus/dactylus, carpus/propodus and coxa/basis subdivisions (*Figure 9Q*), and the carpus/merus subdivision (*Figure 9R*).

## Expression of limb patterning genes validates cellular models of *Parhyale* limb morphogenesis

To test our cellular models and make a first link between expression of limb patterning genes and morphogenetic cell behaviors, we analyzed by in situ hybridization the expression of the *Parhyale decapentaplegic* (*Ph-dpp*) gene that encodes a Bone Morphogenetic Protein 2/4 signaling molecule (*Figure 10—figure supplement 1G*). In *Drosophila*, Dpp signaling controls dorsal cell fate in the leg and growth via cell proliferation in the wing (*Barrio and Milán, 2017*; *Bosch et al., 2017*; *Brook and Cohen, 1996*; *Matsuda and Affolter, 2017*; *Rogulja and Irvine, 2005*; *Svendsen et al., 2015*). Therefore, probing *Ph-dpp* expression in *Parhyale* limb buds could provide a direct test for our cell-based predictions regarding the DV lineage restriction and the differential cell proliferation rates in the limb primordium.

Analysis of embryos 84–96 hr AEL revealed alternating regions of high/moderate and low/no *Ph-dpp* expression in the anterior thoracic region (*Figure 10A,A'* and *Figure 10—figure supplement 1A,A'*). We used MaMuT to annotate both the gene expression and identity of cells in stained T2 limbs at cellular resolution. Acknowledging that the graded *Ph-dpp* expression obscured the precise limits of its expression, this analysis suggested that the region of high/moderate *Ph-dpp* expression was localized to rows E4c, E4d and E5a that mostly contribute to the presumptive dorsal compartment, while low/no *Ph-dpp* expression could be detected in the prospective ventral rows E4b

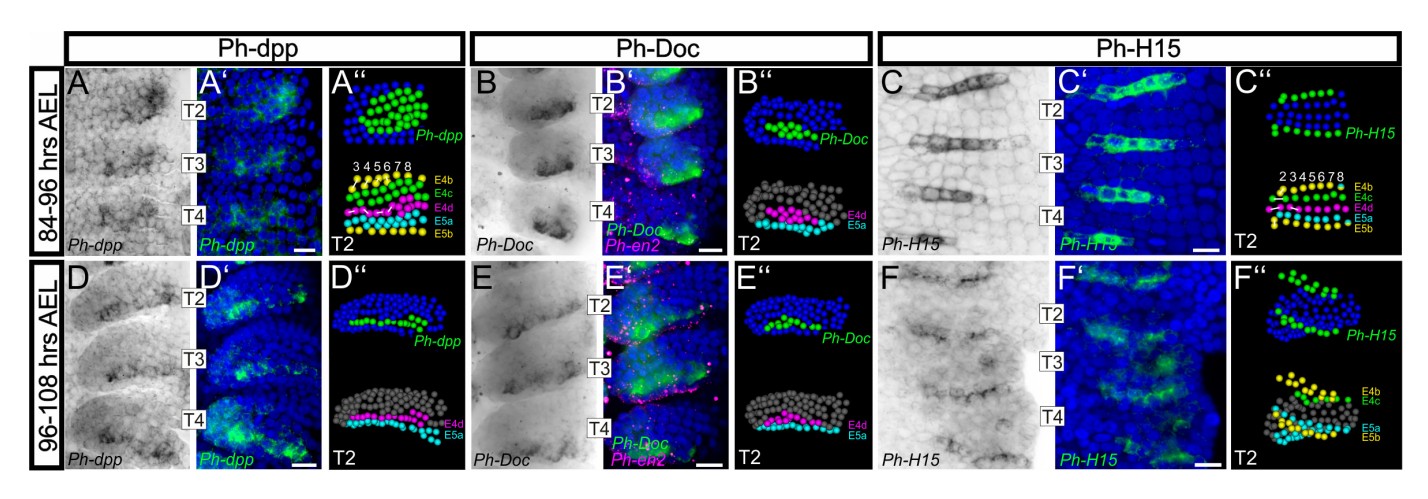

**Figure 10.** Analysis of developmental regulatory genes corroborates cellular models of limb morphogenesis (see also *Figure 10—figure supplement 1*). (A–F) Brightfield images of T2, T3 and T4 limbs from S16-S18 embryos (top row, 84–96 hr AEL) and S19 embryos (bottom row, 96–108 hr AEL) stained by in situ hybridization for *Ph-dpp* (left columns), *Ph-Doc* (middle columns) and *Ph-H15* (right columns). (A'–F') Same limbs as in panels A–F with the nuclear DAPI staining in blue overlaid with the *Ph-dpp*, *Ph-Doc* or *Ph-H15* pattern false-colored in green. Embryos stained for *Ph-Doc* were co-hybridized with *Ph-en2* shown in magenta to label the posterior compartment. (A''–F'') MaMuT reconstructions of the T2 limbs shown in panels A–F. The top panels are color-coded by gene expression with *Ph-dpp*, *Ph-Doc* or *Ph-H15* expressing cells shown in green and non-expressing cells in blue. Bottom panels indicate the identity of the same cells; cells are color-coded by AP rows, column number is shown at the top and white lines connect sister cells. All panels show ventral views with anterior to the top and ventral midline to the left. Scale bars are 20 μm.
DOI: https://doi.org/10.7554/eLife.34410.024

The following figure supplement is available for figure 10:

**Figure supplement 1.** Expression of *Ph-dpp*, *Ph-Doc* and *Ph-H15* during *Parhyale* limb bud formation.
DOI: https://doi.org/10.7554/eLife.34410.025

anteriorly and E5b posteriorly (*Figure 10A''*). Furthermore, *Ph-dpp* expression faded in the medial (prospective ventral) columns and the border between high/moderate and low/no expressing cells was located in descendent cells from column 4 as also predicted by our in silico cellular analysis (*Figure 10A–A''*). In embryos 96–108 hr AEL, the domain of strong *Ph-dpp* expression was more localized in the row of anterior-dorsal cells abutting the AP boundary (*Figure 10D–D''* and *Figure 10—figure supplement 1D,D'*).

To get an insight into the downstream effects of Dpp signaling in the *Parhyale* limb, we also analyzed expression of the Tbx6/*Dorsocross* (*Doc*) gene (*Figure 10—figure supplement 1H*) that responds to high levels of Dpp signaling in the dorsal region of the *Drosophila* embryo and leg disc (*Svendsen et al., 2015*). Expression of the single *Doc* gene identified in *Parhyale* (*Ph-Doc*) was detected in a subset of the *Ph-dpp*-expressing cells at 84–96 hr AEL (*Figure 10B–B''* and *Figure 10—figure supplement 1B,B'*), while 12 hr later the two genes exhibited essentially identical strong expression in the cells abutting the AP boundary (*Figure 10E–E''* and *Figure 10—figure supplement 1E,E'*). In both stages analyzed, cells expressing *Ph-dpp* and *Ph-Doc* also exhibited the highest rates of cell proliferation (compare *Figure 10D'',E''* with *Figure 6G,H*) providing strong correlative evidence for a morphogen-dependent control of *Parhyale* limb growth.

As a last validation of our cellular models, we probed the expression of the *Parhyale H15* (*Ph-H15*) gene during early limb formation (*Figure 10—figure supplement 1H*). In *Drosophila* and other arthropods studied, the Tbx20 genes *H15/midline* act antagonistically to dorsal selector genes and control ventral cell fate in developing legs (*Janssen et al., 2008*; *Svendsen et al., 2015*). Our model for the timing and position of limb DV compartmentalization predicted that *Ph-H15* would come up in the b cells from the 4-row-parasegment stage onwards. In agreement with these predictions, in situ hybridization analyses detected the *Ph-H15* transcripts specifically in the b row cells. Furthermore, expression initiated shortly after the ab cells divided longitudinally into the a and b daughter cells in each forming 4-row-parasegment (*Figure 10C–C''* and *Figure 10—figure supplement 1C, C'*). Although *Ph-H15* was first activated in all b cells, during later divisions *Ph-H15* expression faded in the more medial columns (*Figure 10C–C''*) and persisted only in the ventral limb cells close to the body wall (*Figure 10F–F''* and *Figure 10—figure supplement 1F,F'*).

All these results demonstrated how the reconstruction of cell lineages and behaviors can provide solid predictions and powerful contexts to study the expression and function of associated genes.

## Discussion

We have established an integrated framework to study the cellular and genetic basis of developmental morphogenesis. By combining light-sheet microscopy with new software for cell tracking in large multi-dimensional datasets, we have revealed the cellular architecture and dynamics underlying epithelial remodeling and organ morphogenesis in a non-conventional experimental model.

### Reconstruction of *Parhyale* embryogenesis with multi-view LSFM and MaMuT

The LSFM technology is empowering biologists to image developmental processes with unprecedented spatiotemporal resolution. Together with MaMuT-based lineaging and tracking, various experimental designs can be addressed ranging from analyzing a small subset of objects in the imaged volume to systems-wide analyses of all constituent parts.

The lineage reconstructions presented in this article were generated manually and required 2 to 3 months for each limb. More generally, manual lineaging efforts can take anything between few days to several months depending on the number of tracked cells, the complexity of the imaged tissue of interest, the duration of the tracked process, the quality of the image dataset, and the desired accuracy and completeness of the reconstructed lineages. The main advantage of manual tracking by experts is that the extracted lineage is more likely error-free compared to results of automated trackers that must be manually proofread before any meaningful analysis can be attempted. In addition to allowing reliable biological insights, manually generated lineages serve as important 'ground truth' datasets for the application of machine learning based automated tracking solutions (*Ulman et al., 2017*).

Acknowledging that fully manual tracking is a laborious and repetitive task that may be impractical for large-scale comparative lineaging approaches, the latest MaMuT architecture offers, in

addition to manual tracking, two functionalities for automated tracking: (i) a semi-automated option where individual nuclei can be selected by the user and tracked computationally over time, and (ii) the option to import into MaMuT fully automated annotations generated by the Tracking with Gaussian Mixture Models (TGMM) software (*Amat et al., 2014*), which is currently one of the most accurate and computationally efficient methods for segmentation and tracking of fluorescently labeled nuclei. After the import, MaMuT can be used to manually proofread and correct the results of the automated tracking pipeline. However, we also note that the graph data structure in MaMuT can handle efficiently up to about a hundred thousand annotations. This number is well within the realm of manually generated annotations, but is normally exceeded by large-scale fully automated lineaging engines like TGMM. As a trade-off until this constraint is addressed in the future, we also provide users the option to crop the imported TGMM annotations in space and/or in time to make them compatible with MaMuT.

The crustacean *Parhyale* is already an attractive new model for developmental genetic and functional genomic studies (*Kao et al., 2016*; *Liubicich et al., 2009*; *Martin et al., 2016*; *Pavlopoulos et al., 2009*; *Stamataki and Pavlopoulos, 2016*). By extending here the experimental toolkit with multi-view LSFM and cellular reconstructions with MaMuT, it is feasible to study gene expression and function in the context of single-cell resolution fate maps. Especially when it comes to appendage development, the *Parhyale* body plan provides exceptional material to probe the molecular and cellular basis of tissue patterning, growth and differentiation during normal embryogenesis and post-embryonic regeneration (*Alwes et al., 2016*; *Konstantinides and Averof, 2014*).

The tempo and mode of development has also important ramifications for *Parhyale* imaging and tracking. The relatively slow tempo of development enables us to image embryos at a very high spatial resolution through the acquisition of multiple and highly overlapping views without compromising the temporal resolution. *Parhyale* can match the spatiotemporal resolution of *Drosophila* or zebrafish LSFM datasets, even when access to highest-speed instruments is not available. Due to the optical clarity of the embryo and positioning of the appendages on the surface of the developing embryo, all constituent cells can be followed for quantitative analyses. Finally, the stereotyped and ordered organization of the *Parhyale* ectoderm will allow to identify homologous cells and compare lineages, cell behaviors and associated genes between serially homologous structures in the same embryo, across embryos and even across malacostracan crustaceans (*Browne et al., 2005*; *Dohle et al., 2004*; *Dohle and Scholtz, 1988*; *Gerberding et al., 2002*; *Hejnol and Scholtz, 2004*; *Scholtz, 1990*; *Scholtz et al., 1994*; *Wolff and Gerberding, 2015*; *Wolff and Scholtz, 2002*; *2008*).

## Cellular basis of arthropod limb morphogenesis: lessons from *Parhyale*

The combination of multi-view light-sheet imaging and tracking has enabled a detailed analysis of the dynamics of all constituent cells in an outgrowing and elongating animal limb. So far, these descriptions have been only partly available for *Drosophila* limbs that are derived and not representative for many insects, much less arthropods in general, in two very important respects. First, limb specification, patterning, growth and differentiation take place at distinct developmental stages during embryonic, larval and pupal development. On the contrary, all these processes come about during embryogenesis in most other arthropods, including *Parhyale*. In addition to these heterochronic shifts, limb patterning mechanisms in *Drosophila* operate in the flat imaginal disc epithelia, rather than the 3D epithelial outgrowths observed in *Parhyale* that are typical for most other arthropod limbs.

Classical lineaging experiments revealed that tissue compartmentalizations in the *Drosophila* wing and leg primordia take place along the AP axis during early embryogenesis and along the DV axis during larval development (*Garcia-Bellido et al., 1973*; *Steiner, 1976*). Our understanding of the AP and DV organization in other arthropod limbs has relied so far entirely on gene expression studies. Expression of segment polarity genes, like *en* and *wingless* (*wg*), has demonstrated that the AP separation is conserved across arthropods and takes place during segmentation (*Angelini and Kaufman, 2005*; *Damen, 2007*). In *Parhyale*, the AP compartment boundary is established at the 1-row stage at the interface of neighboring parasegments (*Browne et al., 2005*; *Dohle and Scholtz, 1988*; *Hejnol and Scholtz, 2004*; *Scholtz et al., 1994*). With the exception of descriptive gene expression studies (*Angelini and Kaufman, 2005*; *Damen, 2007*; *Janssen et al., 2008*), the mechanism, timing and position of the DV separation in arthropod limbs has remained unexplored at the

cellular level due to the lack of lineage tracing methodologies. Even in *Drosophila*, it is not entirely clear yet whether DV separation in the leg disc relies on heritable or non-heritable subdivisions or a combination of both mechanisms (*Brook and Cohen, 1996*; *Steiner, 1976*; *Svendsen et al., 2015*).

By analyzing the dynamics of digital clones in reconstructed T2 limbs, we have been able to explore the cellular basis of limb patterning in *Parhyale*. This approach first confirmed the position and timing of the known AP compartment boundary, and then revealed a putative heritable subdivision along the DV axis from the 4-row-parasegment stage onwards. Interestingly, expression of the *Distal-less* gene, which is an early marker of limb specification, is first detected at the 4-row-parasegment in the d3/d4 cells located at the intersection of the AP and DV boundaries (*Browne et al., 2005*; *Hejnol and Scholtz, 2004*). This intersection also marks the tip of the forming limb throughout epithelial remodeling and outgrowth. Thus, the *Parhyale* limb appears to perfectly conform to Meihardt's boundary model (*Meinhardt, 1983*). This model postulates that a secondary developmental field, i.e. the PD axis of a limb that is specified during embryogenesis de novo relative to the main AP and DV body axes, initiates and is patterned around the intersection of the AP and DV compartment boundaries.

The inference of the four constituent compartments provided a powerful framework to interpret the cell behaviors during limb development both in a qualitative and quantitative manner. This analysis strongly suggested that a combination of cellular mechanisms is at work to remodel the embryonic epithelium during limb outgrowth. First, there was a significant difference in cell proliferation rates between the center (faster dividing) and the periphery (slower dividing) of the limb primordium from early specification until limb bud formation. Such a growth-based morphogenesis model has been the dominant hypothesis for almost 50 years to explain the outgrowth of the vertebrate limb (*Ede and Law, 1969*; *Hornbruch and Wolpert, 1970*; *Morishita and Iwasa, 2008*; *Searls and Janners, 1971*) – oriented cell motion and division were also recently involved (*Boehm et al., 2010*; *Wyngaarden et al., 2010*) - but has never been implicated as the driving mechanism behind arthropod limb evagination. Limb bud formation can be reduced by inhibiting cell proliferation pharmacologically, as has been demonstrated in larvae of another crustacean with direct developing limbs, the brine shrimp *Artemia* (*Freeman et al., 1992*). Second, limb elongation was tightly associated - and presumably effected - by two patterned cell behaviors: i) increased cell proliferation at the tip of the limb resembling a putative growth zone which generates many of the new cells necessary for limb outgrowth, and ii) strong bias in the orientation of mitotic divisions parallel to the PD axis of growth. Third, the different PD domains of the *Parhyale* limb could be traced back to distinct mediolateral positions in the early germband stage. During limb bud formation and elongation, there was a transition and refinement of these positional values along the PD axis. Fourth, besides the early AP and DV lineage restrictions, we observed a secondary PD separation between neighboring segments during limb segmentation.

Overall, our approach demonstrates that the comprehensive fine-scale reconstruction of a developmental process can shed light into functionally interdependent patterning mechanisms operating across multiple scales.

## Reconciling genetic with cellular models of limb morphogenesis

In the *Drosophila* leg disc, the Dpp and Wg ligands are induced at the AP boundary in the dorsal and ventral cells, respectively. Dpp and Wg create a concentration gradient with the highest overlap in their expression in the center of the disc and cooperate in the establishment of concentric domains of gene expression of a set of limb gap genes that pattern the PD axis (*Estella et al., 2012*). Dpp and Wg signaling also act antagonistically to control dorsal and ventral cell fate through regulation of the downstream selector T-box genes *optomotor blind/Doc* dorsally and *H15/midline* ventrally (*Svendsen et al., 2015*).

The PD expression of the limb gap genes is conserved in arthropods, including *Parhyale* (*Angelini and Kaufman, 2005*; *Browne et al., 2005*; *Prpic and Telford, 2008*). Our analysis of *dpp*, *Doc* and *H15* expression in a crustacean species also suggests conserved roles for these genes in dorsal and ventral cell fate specification, and provides extra independent support for a compartment-based mechanism to pattern the DV axis of arthropod limbs. *Wg* expression is currently not known in *Parhyale*. If it is expressed in a complementary pattern to *Ph-dpp* in the prospective ventral territory, it could point to a similar logic for patterning the limb PD axis like in *Drosophila*. In fact, our reconstructions have suggested that the distal DV margin (that in this scenario would experience

the highest levels of Dpp and Wg signaling) is located between descendent cells from columns 4 and 5. These are indeed the cells that contribute to the distal-most limb segments.

Although the function of the Dpp morphogen gradient in patterning the *Drosophila* limbs is well understood, its role in promoting growth is still controversial (*Akiyama and Gibson, 2015*; *Barrio and Milán, 2017*; *Bosch et al., 2017*; *Harmansa et al., 2015*; *Matsuda and Affolter, 2017*; *Restrepo et al., 2014*; *Rogulja and Irvine, 2005*). The anterior-dorsal cells expressing *Ph-dpp* and *Ph-Doc* were among the fastest dividing cells in the center of the limb primordium. Later, strong expression of *Ph-dpp* and *Ph-Doc* resolved into a row of cells abutting the AP compartment boundary. Again, these cells displayed some of the highest proliferation rates quantified during limb outgrowth, suggesting a Dpp-dependent control of *Parhyale* limb growth. We anticipate that the LSFM imaging and tracking approaches described here, together with the recent application of CRSIPR/Cas-based methodologies for genome editing (*Kao et al., 2016*) will provide excellent tools to further explore how morphogens like Dpp regulate growth and form at cellular resolution.

# Materials and methods

## Key resources table

| Reagent type (species) or resource | Designation | Source or reference | Identifiers | Additional information |
|---|---|---|---|---|
| Strain, strain background (*Parhyale hawaiensis*) | Wild Type | PMID: 15986449 | | |
| Strain, strain background (*P. hawaiensis*) | *PhHS>H2B-mRFPruby* | This paper | | |
| Recombinant DNA reagent | pMi{3xP3>EGFP; PhHS>H2B-mRFPruby} | This paper | | |
| Software, algorithm | MaMuT | This paper | | http://imagej.net/MaMuT |
| Software, algorithm | SIMI°BioCell | PMID: 9133433 | | http://simi.com/en/products/cell-research |
| Gene (*P. hawaiensis*) | *Ph-dpp* | This paper | GenBank: KY696711 | |
| Gene (*P. hawaiensis*) | *Ph-Doc* | This paper | GenBank: KY696712 | |
| Gene (*P. hawaiensis*) | *Ph-en2* | This paper | GenBank: KY696713 | |
| Gene (*P. hawaiensis*) | *Ph-H15* | This paper | | |

## Generation of transgenic *Parhyale* labeled with H2B-mRFPruby

*Parhyale hawaiensis* (*Dana, 1853*) rearing, embryo collection, microinjection and generation of transgenic lines were carried out as previously described (*Kontarakis and Pavlopoulos, 2014*). To fluorescently label the chromatin in transgenic *Parhyale*, we fused the coding sequences of the *Drosophila* histone *H2B* and the *mRFPruby* monomeric Red Fluorescent Protein and placed them under control of a strong *Parhyale* heat-inducible promoter (*Pavlopoulos et al., 2009*). *H2B* was amplified from genomic DNA with primers Dmel_H2B_F_NcoI (5'-TTAACCATGGCTCCGAAAAC TAGTGGAAAG-3') and Dmel_H2B_R_XhoI (5'-ACTTCTCGAGTTTAGAGCTGGTGTACTTGG-3'), and *mRFPruby* was amplified from plasmid pH2B-mRFPruby (*Fischer et al., 2006*) with primers mRFPruby_F_XhoI (5'-ACAACTCGAGATGGGCAAGCTTACC-3') and mRFPruby_R_PspMOI (5'-TA TTGGGCCCTTAGGATCCAGCGCCTGTGC-3'). The NcoI/XhoI-digested H2B and XhoI/PspOMI-digested mRFPruby fragments were cloned in a triple-fragment ligation into NcoI/NotI-digested vector pSL-PhHS>DsRed, placing *H2B-mRFPruby* under control of the *PhHS* promoter (*Pavlopoulos et al., 2009*). The *PhHS>H2B-mRFPruby-SV40polyA* cassette was then excised as an AscI fragment and cloned into the AscI-digested pMinos{3xP3>EGFP} vector (*Pavlopoulos and Averof, 2005*; *Pavlopoulos et al., 2004*), generating plasmid pMi{3xP3>EGFP; PhHS>H2B-mRFPruby}. Three independent transgenic lines were established with this construct for heat-inducible expression of H2B-mRFPruby. The most strongly expressing line was selected for all applications. In this line, nuclear H2B-mRFPruby fluorescence plateaued about 12 hr after heat-shock and

high levels of fluorescence persisted for at least 24 hr post heat-shock labeling chromatin in all cells throughout the cell cycle.

## Multi-view LSFM imaging of *Parhyale* embryos

Standard procedures for multi-view LSFM recordings of *Parhyale* embryogenesis were established after imaging several dozen embryos individually in pilot experiments, first on a Zeiss prototype and, later on, on the commercial Zeiss Lightsheet Z.1 microscope. Several parameters described below were optimized to ensure that the two embryos used for lineage reconstruction (i) survived the recording process and hatched into juveniles without any morphological abnormalities, and (ii) were imaged with the appropriate spatiotemporal resolution and signal-to-noise ratio for accurate and comprehensive cell tracking in developing appendages.

To prepare embryos for LSFM imaging, 2.5 day old transgenic embryos (early germband stage; S11 according to (*Browne et al., 2005*)) were heat-shocked for 1 hr at 37°C. About 12 hr later (stage S13), they were mounted individually in a cylinder of 1% low melting agarose (SeaPlaque, Lonza) inside a glass capillary (#701902, Brand GmbH) with their AP axis aligned parallel to the capillary. A 1:4000 dilution of red fluorescent beads (#F-Y050 microspheres, Estapor Merck) were included in the agarose as fiducial markers for multi-view reconstruction. During imaging, the embedded embryo was extruded from the capillary into the chamber filled with artificial seawater supplemented with antibiotics and antimycotics (FASWA; (*Kontarakis and Pavlopoulos, 2014*)). The FASWA in the chamber was replaced every 12 hr after each heat shock (see below). The Zeiss Lightsheet Z.1 microscope was equipped with a 20x/1.0 Plan Apochromat immersion detection objective and two 10x/0.2 air illumination objectives producing two light-sheets 5.1 μm thick at the waist and 10.2 μm thick at the edges of a 488 μm x 488 μm field of view.

We started imaging *Parhyale* embryogenesis from three angles/views (the ventral side and the two ventral-lateral sides 45° apart from ventral view) during 3 to 4.5 days AEL to avoid photo-damaging the dorsal thin extra-embryonic tissue, and continued imaging from five views (adding the two lateral sides 90° apart from ventral view) during 4.5 to 8 days AEL. A multi-view acquisition was made every 7.5 min at 26°C. The H2B-mRFPruby fluorescence levels were replenished regularly every 12 hr by raising the temperature in the chamber from 26°C to 37°C and heat-shocking the embryo for 1 hr. Each view (z-stack) was composed of 250 16-bit frames with voxel size 0.254 μm x 0.254 μm x 1 μm. Each 1920x1920 pixel frame was acquired using two pivoting light-sheets to achieve a more homogeneous illumination and reduced image distortions caused by light scattering and absorption across the field of view. Each optical slice was acquired with a 561 nm laser and exposure time of 50 msec. With these conditions, *Parhyale* embryos, like the one bearing the T2 limb#1 analyzed in detail with MaMuT, were imaged routinely for a minimum of 4 days or even up to hatching. After hatching, the morphology of imaged specimen was compared between the left and the right side, as well as to its non-imaged siblings, to confirm that no obvious developmental or morphological abnormalities were detected.

The embryo bearing the T2 limb#2 was imaged on a Zeiss LSFM prototype (*Preibisch et al., 2010*) that offered single-sided illumination and single-sided detection with a 40x/0.8 immersion objective. One side of this embryo was imaged from 3 views 40° apart (ventral, ventral-left and left) every 7.5 min over a period of 66 hr. Each view was composed of 150 frames (1388 × 1080 pixels) with voxel size 0.366 μm x 0.366 μm x 2 μm. The embryo was imaged at 29–30°C and was heat-shocked for 1 hr twice a day by perfusing warm FASWA at 37 °C. Cell tracking was carried out with the SIMI°BioCell software (*Hejnol and Scholtz, 2004*; *Schnabel et al., 1997*) on a single view, the ventral-left view, of this dataset. Lineage reconstruction of limb#2 with SIMI°BioCell was complete up to about 22 hr of imaging time (35 hr when scaled to the growth rate of limb#1). After this time-point, an increasing number of cells in limb#2, in particular the descendant cells from the medial columns, became intractable.

## 4d reconstruction of *Parhyale* embryogenesis from multi-view LSFM image datasets

*Parhyale* LSFM acquisitions typically resulted in 192 time-points/240K images/1.7 TB of raw data per day. Image processing was carried out on a MS Windows 7 Professional 64-bit workstation with 2 Intel Xeon E5-2687W processors, 256 GB RAM (16 X DIMMs 16384 MB 1600 MHz ECC DDR3), 4.8

TB hard disk space (2 × 480 GB and 6 × 960 GB Crucial M500 SATA 6 Gb/s SSD), 2 NVIDIA Quadro K4000 graphics cards (3 GB GDDR5). The workstation was connected through a 10 GB network interface to a MS Windows 2008 Server with 2 Intel Xeon E5-2680 processors, 196 GB RAM (24 X DIMM 8192 MB 1600 MHz ECC DDR3) and 144 TB hard disk space (36 X Seagate Constellation ES.3 4000 GB 7200 RPM 128 MB Cache SAS 6.0 Gb/s). All major LSFM image data processing steps were done with software modules available through the Multiview Reconstruction Fiji plugin (http://imagej.net/Multiview-Reconstruction) according to the following steps:

1. Preprocessing: Image data acquired on Zeiss Lightsheet Z.1 were saved as an array of czi files labeled with ascending indices, where each file represented one view (z-stack). czi files were first renamed into the 'spim_TL{t}_Angle{a}.czi' filename, where t represented the time-point (e.g. 1 to 192 for a 1 day recording) and a the angle (e.g. 0 for left view, 45 for ventral-left view, 90 for ventral view, 135 for ventral-right view and 180 for right view), and then resaved as tif files.
2. Bead-based spatial multi-view registration: In each time-point, all views were aligned to each-other and to an arbitrary reference view fixed in 3D space (e.g. views 0, 45, 90, 135 aligned to 180) using the bead-based registration option (*Preibisch et al., 2010*). In each view, fluorescent beads scattered in the agarose were segmented with the Difference-of-Gaussian algorithm using a sigma value of 3 and an intensity threshold of 0.005. Corresponding beads were identified between views and were used to determine the affine transformation model that matched the views within each time-point.
3. Fusion by multi-view deconvolution: Spatially registered views were down-sampled twice for time and memory efficient computations during the image fusion step. Input views were then fused into a single output 3D image with a more isotropic resolution using the Fiji plugin for multi-view deconvolution estimated from the point spread function of the fluorescent beads (*Preibisch et al., 2014*). The same cropping area containing the entire imaged volume was selected for all time-points. In each time-point, the deconvolved fused image was calculated on GPU in blocks of 256 × 256×256 pixels with 7 iterations of the Efficient Bayesian method regularized with a Tikhonov parameter of 0.0006.
4. Bead-based temporal registration: To correct for small drifts of the embryo over the extended imaging periods (e.g. due to agarose instabilities), we stabilized the fused volume over time using the segmented beads (sigma = 1.8 and intensity threshold = 0.005) for temporal registration with the affine transformation model using an all-to-all matching within a sliding window of 5 time-points.
5. Computation of spatiotemporally registered fused volumes: Using the temporal registration parameters, we generated a stabilized time-series of the fused deconvolved 3D images.
6. 4D rendering: The *Parhyale* embryo was rendered over time from the spatiotemporally registered fused data using Fiji's 3D Viewer.

## Lineage reconstruction with the Massive Multi-view Tracker (MaMuT)

MaMuT was developed as a tool for cell lineaging in multi-view LSFM image volumes by enabling to track objects synergistically from all available views. This functionality has a number of advantages. Raw views do not have to be fused into a single volume, which is computationally by far the most demanding step (*Preibisch et al., 2014*). The users also preserve the original redundancy of the data, which in many cases like in *Parhyale* allows capturing cells from two or more neighboring views that can be interpreted independently for a more accurate analysis. Finally, MaMuT allows users to analyze sub-optimal datasets that cannot be fused properly or may create fusion artifacts. Of course, combining the raw views with a high-quality fused volume is the best available option, especially when handling complex datasets with high cell densities.

While offering multi-view tracking, MaMuT delivers also other important functionalities. First, it is a turnkey software solution with a convenient interface for interactive exploration, annotation and curation of image data. Any image acquired by any microscopy modality that can be opened in Fiji can be also imported into MaMuT. Second, MaMuT offers a highly responsive and interactive navigation through multi-terabyte datasets. Individual z-stacks representing different views, channels and time-points of a multi-dimensional dataset can be displayed independently or in combinations in multiple synced Viewer windows. Third, objects of interest like cells and nuclei (spots) can be selected synergistically from all available Viewers and followed over time to reconstruct their trajectories (tracks) and lineage information. Fourth, the created spots and tracks can be visualized and

edited interactively in the Viewers and the TrackScheme lineage browser, and animated in the 3D Viewer. For visual interpretation of the data, annotations can be colored based on the primary lineage information or derived numerical parameters. Fifth, lineages can be reconstructed in a manual, semi-automated or fully automated manner followed by manual curation if necessary. Sixth, all spot and track information can be exported from MaMuT to other interfaces for more specialized analyses. Seventh, decentralized annotation by multiple users has been made possible by also developing a web service for remote access to large image volumes stored online. Following on the tradition of the Fiji community for open-source distribution of biological image analysis software, MaMuT is provided freely and openly to the community, it is extensively documented and can be customized by other users.

In practical terms, for lineaging purposes, the *Parhyale* multi-view LSFM raw views were registered spatiotemporally and the image data together with the registration parameters were converted into the custom HDF5/XML file formats utilized by the BigDataViewer and MaMuT Fiji plugins. The MaMuT reconstruction of the *Parhyale* T2 limb described in this article required about 10 weeks of dedicated manual cell tracking by an experienced annotator. The raw image data were displayed in Viewer windows and each z-stack was visualized in any desired color and brightness, scale (zoom), translation (position) and rotation (orientation). All Viewer windows were synced based on the calculated registration parameters and shared a common physical coordinate system; upon selecting an object of interest (spot) in one Viewer, the same spot was identified and displayed in all other windows, and its x, y, z position was mapped onto this common physical space. To guarantee the accuracy of our lineage reconstructions, the center of each tracked nucleus was verified in at least two neighboring views and by slicing the data orthogonally in separate Viewer windows. The nuclei contributing to the T2 limb of interest were identified in the first time-point and tracked manually every five time-points except during mitosis, in which case we also tracked one time-point before and one after segregation of the daughter chromosomes during anaphase/telophase. The reconstructed trajectories and lineages were also displayed in two additional synced windows, the TrackScheme and the 3D Viewer. The TrackScheme lineage browser and editor displayed the reconstructed cell lineage tree with tracked nuclei represented as nodes connected by edges over time and cell divisions depicted as split branches in the tree. The 3D Viewer window displayed interactive animations of the spots depicted as spheres and their tracks over time. The spots and the tracks in the Viewer, TrackScheme and 3D Viewer windows could be color-coded by lineage, position and other numerical features to assist visual analysis and interpretation of the data. In addition, all these windows were synced to simultaneously highlight active spots of interest at the selected time-point, greatly facilitating the cell lineaging process.

## Comparison of reconstructed lineage trees

For comparative purposes, each reconstructed lineage tree was defined as a set of division times. For example, let's consider a lineage tree $L$ that starts with cell $d$. Cell $d$ divides at time $t_0$ giving rise to the two daughter cells $d_1$ and $d_2$. Then $d_1$ divides at time $t_1$ giving rise to daughter cells $d_{11}$ and $d_{12}$. Finally, $d_{12}$ divides at time $t_{12}$ giving the daughter cells $d_{121}$ and $d_{122}$. In this scenario, we define $L$ as $L = \{t_0, t_1, t_{12}\}$.

Let's now consider two lineage trees $L^x$ and $L^y$, where x and y refer to the founder cells whose lineage trees are under comparison (e.g. x corresponds to E4c5 cell from limb#1 and y to E5b6 cell from limb#2). In order to be comparable, these two lineage trees need to be registered temporally. In our study, we performed a linear rescaling by an empirically determined factor of 1.6 to match the increase in cell number between limb#1 and limb#2 that were imaged at different temperatures and exhibited different growth rates.

We then defined $\Delta(L^x, L^y)$ as the distance between the two registered lineage trees. This distance takes into consideration two metrics, the difference in the timing of divisions and the difference in the number of divisions between the two lineages, and is computed in the following way:

$$\Delta(L^x, L^y) = \frac{\delta_t(L^x, L^y)/n_t + \delta_n(L^x, L^y)/n_n}{2}$$

In this equation, $\delta_t(L^x, L^y)$ is the difference in the timing of divisions and $\delta_n(L^x, L^y)$ is the difference in the number of division between the two lineages. $n_t$ and $n_n$ are used to normalize the two

metrics so that their values are comparable. They are defined as the maximum values observed for $\delta_t(L^x, L^y)$ and $\delta_n(L^x, L^y)$ in a given run of pairwise comparisons, i.e. they are the maximum values obtained in the $34 \times 34$ comparisons to calculate the distances between the 34 founder cells within limb#1 or within limb#2 or between limb#1 and limb#2.

$\delta_n$ is computed as the absolute value of the difference between the respective numbers of divisions in the two lineage trees:

$$\delta_n(L^x, L^y) = |Card(L^x) - Card(L^y)|$$

To calculate $\delta_t$, we first paired the division times between the two lineage trees. For such a pairing $P = \left\{ \left( t_i^x, t_j^y \right) \mid t_i^x \in L^x, \ t_j^y \in L^y \right\}$ the difference in division times $\delta_t(P)$ is computed as follows:

$$\delta_t(P) = \frac{1}{Card(P)} \sum_{(t_i^x, t_j^y) \in P} |t_i^x - t_j^y|$$

The pairing $P^\star$ that minimizes $\delta_t$ is used to compute the temporal distance between the lineage trees. Let $P$ be the set of all possible pairings, then $P^\star$ is defined as followed:

$$P^\star = \underset{P \in P}{\operatorname{argmin}} \, \delta_t(P)$$

We then define $\delta_t$ as $\delta_t = \delta_t(P^\star)$.

Once we computed all the pairwise distances between lineages of the cells under comparison, hierarchical clustering was performed using Ward's method. For the hierarchical clustering in the average *Parhyale* T2 limb, we combined for each founder cell the information from the two limbs. The average lineage tree $L_{12}^x$ of lineage trees $L_1^x = \{t_{11}^x, t_{12}^x, t_{13}^x\}$ and $L_2^x = \{t_{21}^x, t_{22}^x, t_{23}^x, t_{24}^x\}$, where x corresponds to the founder cell x with lineage trees $L_1^x$ in limb#1 and $L_2^x$ in limb#2, is defined as $L_{12}^x = L_1^x \cup L_2^x = \{t_{11}^x, t_{12}^x, t_{13}^x, t_{21}^x, t_{22}^x, t_{23}^x, t_{24}^x\}$. The computation of the pairwise distance $\Delta$ between average lineage trees was then performed as described above.

### Analysis of gene expression

*Parhyale decapentaplegic* (*Ph-dpp*), *Dorsocross* (*Ph-Doc*), *engrailed-2* (*Ph-en2*) and *H15* (*Ph-H15*) genes were identified by BLAST analysis against the *Parhyale* transcriptome and genome (*Kao et al., 2016*) using the protein sequence of *Drosophila* orthologs as queries. Sequence accession numbers are KY696711 for *Ph-dpp*, KY696712 for *Ph-Doc*, and KY696713 for *Ph-en2*. Phylogenetic tree construction was performed with RAxML using the WAG + G model from MAFFT multiple sequence alignments trimmed with trimAl (*Stamatakis, 2014*). In situ hybridizations were carried out as previously described (*Rehm et al., 2009*). Stained samples were imaged on a Zeiss 880 confocal microscope using the Plan-Apochromat 10x/0.45 and 20x/0.8 objectives. Images were processed using Fiji and Photoshop CS6 (Adobe Systems Inc). For color overlays, the brightfield image of the *Ph-dpp*, *Ph-Doc* or *Ph-H15* BCIP/NBT staining was inverted, false-colored green and merged with the fluorescent signal of the *Ph-en2* FastRed staining in magenta and the nuclear DAPI signal in blue. In order to map gene expression patterns onto cell lineages, the z-stacks from imaged fixed specimens were imported into MaMuT and the manually reconstructed nuclei and annotated gene expression patterns were compared with the corresponding stages of the live imaged and lineaged embryos. This analysis was performed with single-cell accuracy thanks to the well characterized and invariant patterns of cell division across *Parhyale* embryos, the orderly arrangement of cells in the earlier stages analyzed, and the easily identifiable straight boundary between anterior and posterior cells in the later stages analyzed.

## Acknowledgements

We are grateful to Michael Akam and Gerhard Scholtz for actively supporting the early phases of this project, Stephan Saalfeld for assistance with image rendering, Fernando Amat for his help with importing TGMM annotations into MaMuT, Damian Kao for assistance with phylogenetics, Michalis Averof and Matthew Benton for comments on the manuscript, and Angeliki-Myrto Farmaki for designing the MaMuT logo. CW was supported by the Einstein Foundation Berlin grant A-2012_114.

JYT and SS were supported by intramural funding from the Pasteur Institute and ANR Program Grandes Investissement de l'avenir France BioImaging Consortium. SP was supported by core funding from the Max Delbrück Center. TP and PT were supported by the core funding from Max Planck Society and the European Research Council Community's FP7 grant agreement 260746. ES, LG, PJK and AP were supported by the Howard Hughes Medical Institute. AP was also supported by an EC Marie Skłodowska-Curie fellowship.

## Additional information

### Funding

| Funder | Grant reference number | Author |
| --- | --- | --- |
| Howard Hughes Medical Institute | | Evangelia Stamataki<br>Léo Guignard<br>Philipp J Keller<br>Anastasios Pavlopoulos |
| H2020 Marie Skłodowska-Curie Actions | FP7-IEF 302235 | Anastasios Pavlopoulos |
| Max Planck Institute of Molecular Cell Biology and Genetics | | Tobias Pietzsch<br>Benjamin Harich<br>Pavel Tomancak |
| European Research Council | 260746 | Tobias Pietzsch<br>Pavel Tomancak |
| Einstein Stiftung Berlin | A-2012_114 | Carsten Wolff |
| Institut Pasteur | | Jean-Yves Tinevez<br>Spencer Shorte |
| Agence Nationale de la Recherche | | Jean-Yves Tinevez<br>Spencer Shorte |
| Helmholtz-Gemeinschaft | | Stephan Preibisch |

The funders had no role in study design, data collection and interpretation, or the decision to submit the work for publication.

### Author contributions
Carsten Wolff, Conceptualization, Data curation, Formal analysis, Validation, Visualization, Methodology, Writing—original draft, Writing—review and editing, Performed image analysis and lineage reconstructions; Jean-Yves Tinevez, Tobias Pietzsch, Data curation, Software, Methodology, Writing—review and editing, Developed MaMuT; Evangelia Stamataki, Formal analysis, Validation, Investigation, Visualization, Writing—review and editing, Performed in situ hybridizations; Benjamin Harich, Investigation, Generated image data; Léo Guignard, Software, Formal analysis, Visualization, Performed lineage comparisons; Stephan Preibisch, Data curation, Software, Performed image analysis; Spencer Shorte, Data curation, Funding acquisition; Philipp J Keller, Data curation, Software; Pavel Tomancak, Supervision, Funding acquisition, Methodology, Project administration, Writing—review and editing; Anastasios Pavlopoulos, Conceptualization, Data curation, Formal analysis, Supervision, Funding acquisition, Validation, Investigation, Visualization, Methodology, Writing—original draft, Project administration, Writing—review and editing, Generated transgenic lines and image data, Performed image analysis and lineage reconstructions

### Author ORCIDs
Carsten Wolff http://orcid.org/0000-0002-5926-7338
Philipp J Keller http://orcid.org/0000-0003-2896-4920
Pavel Tomancak http://orcid.org/0000-0002-2222-9370
Anastasios Pavlopoulos http://orcid.org/0000-0002-0230-5815

### Decision letter and Author response
Decision letter https://doi.org/10.7554/eLife.34410.030

Author response https://doi.org/10.7554/eLife.34410.031

## Additional files

### Supplementary files
• Transparent reporting form
DOI: https://doi.org/10.7554/eLife.34410.026

### Major datasets
The following previously published dataset was used:

| Author(s) | Year | Dataset title | Dataset URL | Database, license, and accessibility information |
| --- | --- | --- | --- | --- |
| Kao D, Lai AG, Stamataki E, Rosic S, Konstantinides N, Jarvis E, Di Donfrancesco A, Pouchkina-Stancheva N, Sémon M, Grillo M, Bruce H, Kumar S, Siwanowicz I, Le A, Lemire A, Eisen MB, Extavour C, Browne WE, Wolff C, Averof M, Patel NH, Sarkies P, Pavlopoulos A, Aboobaker A | 2016 | Parhyale hawaiensis isolate: Chicago-F Genome sequencing and assembly | https://www.ncbi.nlm.nih.gov/bioproject/306836 | Publicly available at NCBI BioProject (accession no. PRJNA306836) |

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
