## [Decision Letter]

Thank you for submitting your article "Reconstruction of cell lineages in arthropod limbs with multi-view light-sheet imaging and tracking" for consideration by *eLife*. Your article has been favorably evaluated by Marianne Bronner (Senior Editor) and two reviewers, one of whom, Alejandro Sánchez Alvarado (Reviewer #1), is a member of our Board of Reviewing Editors. The following individual involved in review of your submission has agreed to reveal their identity: Andreas Hejnol (Reviewer #2).

The reviewers have discussed the reviews with one another and the Reviewing Editor has drafted this decision to help you prepare a revised submission.

Summary:

This Tools and Resources paper by Wolff et al., describes the application of a new Fiji plug-in named MaMuT to the study of *Parhyale hawaiensis* embryogenesis. MaMuT is designed to execute cell tracking in multidimensional datasets. In particular, the authors attempt to quantify multiview light-sheet data of the morphogenetic cellular behaviors operating during the direct development of arthropod limbs.

The reviewers agree that this effort is of great interest to the community of developmental biologists interested in real-time imaging quantification. More importantly, the fact that the software is open source will allow the community to put MaMuT through its paces and eventually help determine its range of utility and limitations.

Essential revisions:

Before the manuscript can be further considered, the authors must address the following concerns:

1) The title is not justified. As written this paper is *not* about arthropod limbs in general but about a single arthropod limb in a single embryo of a single species. As such, it is misleading and should be changed to reflect what the paper is really about: a method to do cell tracking.

2) The text of the manuscript tends to be somewhat hyperbolic at places. We believe this is not necessary given the quality of the data presented. An additional shortcoming of doing this is that it prevents the readers to assess the utility of the method as little to no effort is put forward to describe its limitations. We suggest that the authors spend some time to write text in the manuscript discussing the limitations of the method being reported.

3) Although proof-of-principle of the method presented is deftly described in this manuscript, an important question remains unanswered, which is, how much variability there may or may not be between individual specimens? No direct comparison between the two thoracic limbs is reported. Are these different? If so, how different? Given the purpose of cell tracking, it matters both biologically and methodologically speaking whether or not there are cell number differences, positional variability, and stereotypic versus stochastic behaviors. For instance, if the behavior recorded is stereotypic, the authors can provide names of the individual cells that can be probably recognized in both legs and thus provide the first identification of individual cells which then can be used for comparisons between legs. Also, why is there no comparison between the T2 data with a detailed analysis of a pleopod of choice? Pleopods show a different morphology and one could learn about the changes cells undergo to create different morphologies. It seems the data has already been generated and is in the authors' hands. Given that the Discussion contains notable amounts of speculations on how changes in cell patterns can lead to changes in morphology and how gene expression relate to this, why not demonstrate it in this work? In sum, all of these factors impinge upon gene expression pattern interpretations and should be addressed/discussed by the authors.

4) Looking through the data, we had difficulties determining how time-consuming this method may or may not be. This is important to know if, for example, more than one object were to be analyzed or for experiments in which multiple samples need to be measured and compared to each other. The authors could provide the reader with a timeline or with suggestions of how to increase the throughput of this method.

---

## [Author Response]

Essential revisions:Before the manuscript can be further considered, the authors must address the following concerns:1) The title is not justified. As written this paper is not about arthropod limbs in general but about a single arthropod limb in a single embryo of a single species. As such, it is misleading and should be changed to reflect what the paper is really about: a method to do cell tracking.

Our article describes the development of MaMuT for cell tracking and lineage reconstruction in specimens imaged with multi-view light-sheet microscopy. As a first application and proof of principle for MaMuT’s functionalities, we have revealed the cellular architecture and dynamics underlying direct outgrowth of a single limb type in imaged *Parhyale* embryos. Therefore, we appreciate the reviewers’ remark and have modified the title of our article as follows: “Multi-view light-sheet imaging and tracking with the MaMuT software reveals the cell lineage of a direct developing arthropod limb”

2) The text of the manuscript tends to be somewhat hyperbolic at places. We believe this is not necessary given the quality of the data presented. An additional shortcoming of doing this is that it prevents the readers to assess the utility of the method as little to no effort is put forward to describe its limitations. We suggest that the authors spend some time to write text in the manuscript discussing the limitations of the method being reported.

We have addressed the reviewers’ concerns by modifying the text. In this context, we also discuss the limitations of our methodology with regard to the laborious and repetitive nature of manual cell tracking, as well as the partial incompatibility of MaMuT to handle the hundred thousands or millions of annotations that can be generated from automated segmentation and tracking approaches. The paragraph discussing these limitations has been expanded and transferred from Materials and methods to the section “Reconstruction of *Parhyale* embryogenesis with multi-view LSFM and MaMuT” in the Discussion:

“The lineage reconstructions presented in this article were generated manually and required 2 to 3 months for each limb. […] As a trade-off until this constraint is addressed in the future, we also provide users the option to crop the imported TGMM annotations in space and/or in time to make them compatible with MaMuT.”

3) Although proof-of-principle of the method presented is deftly described in this manuscript, an important question remains unanswered, which is, how much variability there may or may not be between individual specimens? No direct comparison between the two thoracic limbs is reported. Are these different? If so, how different? Given the purpose of cell tracking, it matters both biologically and methodologically speaking whether or not there are cell number differences, positional variability, and stereotypic versus stochastic behaviors. For instance, if the behavior recorded is stereotypic, the authors can provide names of the individual cells that can be probably recognized in both legs and thus provide the first identification of individual cells which then can be used for comparisons between legs. Also, why is there no comparison between the T2 data with a detailed analysis of a pleopod of choice? Pleopods show a different morphology and one could learn about the changes cells undergo to create different morphologies. It seems the data has already been generated and is in the authors' hands. Given that the Discussion contains notable amounts of speculations on how changes in cell patterns can lead to changes in morphology and how gene expression relate to this, why not demonstrate it in this work? In sum, all of these factors impinge upon gene expression pattern interpretations and should be addressed/discussed by the authors.

We agree with the reviewers that our article would benefit from the comparison between the two reconstructed T2 limbs. Therefore, we performed such comparison facilitated crucially by Léo Guignard, who has been added to the author list. In particular, i) we compared the birth time of the 34 founder cells in each limb primordium, ii) we compared the increase in cell number as a proxy for the growth rate of each limb, iii) we devised a new metric to compare lineage trees based on the timing and number of divisions, iv) we used this metric to perform a hierarchical clustering of the 34 founder cells in each limb, and v) we used this metric to compare the cell lineages between homologous and non-homologous cells in the two limbs. These analyses are shown in the two new Figures 5 and 7. We have also split former Figure 4 into new Figures 4 and 6 and have renamed old Figures 5 to 7 to new Figures 8 to 10.

In the future, we will advance these comparative approaches to a larger scale and address more adequately key questions regarding the stereotypy/variability of cellular behaviors in growing organs and the cellular basis of serial homology. In order to achieve that, we need to decrease the labor required for each lineage reconstruction from months to weeks, and to develop new computational tools for the systematic comparison and visualization of cell lineage information. Our labs are actively generating the necessary software and analysis algorithms and will share them with the scientific community in a timely manner via the same open access principles deployed for MaMuT.

4) Looking through the data, we had difficulties determining how time-consuming this method may or may not be. This is important to know if, for example, more than one object were to be analyzed or for experiments in which multiple samples need to be measured and compared to each other. The authors could provide the reader with a timeline or with suggestions of how to increase the throughput of this method.

The new text in the Discussion (subsection “Reconstruction of *Parhyale* embryogenesis with multi-view LSFM and MaMuT”, second paragraph) described above in point 2 addresses how time-consuming manual lineaging has been in our case study and what steps have been taken to increase the throughput of lineaging efforts in the future. Besides this text in the Discussion, we have added the following statement in Materials and methods: “The MaMuT reconstruction of the *Parhyale* T2 limb described in this article required about 10 weeks of dedicated manual cell tracking by an experienced annotator.”